# Scalable Circuit Learning for Interpreting Large Language Models

## Abstract

A prominent research direction within mechanistic interpretability involves learning sparse circuits to model causal relationships between LLM components, thereby providing insights into model behavior. However, due to the polysemantic nature of LLM components, learned circuits are often difficult to interpret. While sparse autoencoder (SAE) features enhance interpretability, their high dimensionality presents a significant challenge for existing circuit learning methods to scale. To address these limitations, we propose a scalable circuit learning approach, `CircuitLasso`, that leverages sparse linear regression. Our method can efficiently uncover relationships among SAE features, showing how human-interpretable semantic features propagate through the model and influence its predictions. We empirically evaluate our method against state-of-the-art baselines on benchmark circuit learning tasks, demonstrating substantial improvements in efficiency while accurately capturing circuits involving LLM components. Given its efficiency, we then apply our method to SAE (high dimensional) features and obtain human-interpretable circuits for a grammatical classification task that has not been studied before in mechanistic interpretation. Finally, we validate the utility of our learned circuits by leveraging their insights to improve downstream performance in domain generalization.

## 1 Introduction

The fundamental challenge of mechanistic interpretability is to understand the "why" behind the behaviors of large language models (LLMs). A key technique involves discovering causal circuits, which are compact subgraphs connecting key components within the model (such as attention heads and neurons) that drive a specific behavior or capability. However, existing methods for circuit learning often face a bottleneck. The raw components of an LLM, such as individual neurons, are known to be polysemantic, meaning that a single neuron can be activated by and contribute to multiple, seemingly unrelated concepts. This polysemanticity renders the learned circuits dense, noisy, and challenging for humans to interpret, undermining the very goal of mechanistic interpretability.

The limitations of using raw, polysemantic neurons have motivated a shift toward a more promising foundation for circuit analysis based on Sparse Autoencoders (SAEs) and related tools. SAEs are neural networks trained to reconstruct the activations of an LLM's raw components using a high-dimensional but sparse set of "features". Remarkably, these SAE features tend to be monosemantic, i.e., each feature consistently activates for a single, human-interpretable concept, such as "related to sports," "a specific emotion," or "a particular grammatical structure." The monosemanticity of SAE features has the potential not only to enhance interpretability in itself but also to yield sparser, cleaner causal graphs, and perhaps more faithful representations of the model's internal processing.

Our work is motivated by the above potential of SAE features to achieve greater interpretability in LLM circuit analysis. However, existing circuit learning methods, many of which are designed for the lower-dimensional space of raw neurons, struggle to scale to the high-dimensional feature space of SAEs. The computational complexity and the risk of finding spurious correlations increase dramatically. To address this, we introduce a novel approach to handle the high dimensionality. Our method, `CircuitLasso`, utilizes the Lasso (i.e., $\ell_1$-penalized linear regression) to find a sparse set of connections between features that explains the model's behavior. Sparse linear regression is well-suited for high-dimensional data, as it is computationally efficient and the sparsity translates to

more interpretable circuits. An advantage of our approach is its use of observational data only. This

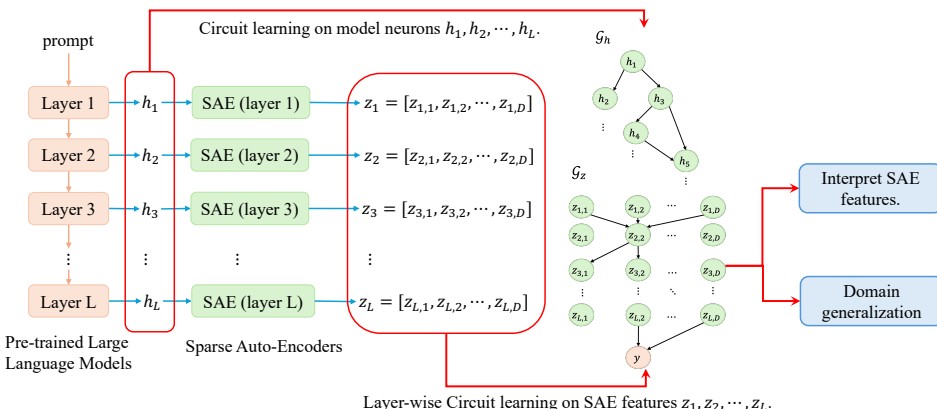

Figure 1: An illustration of our model neuron activation and SAE feature collection procedure, learned circuits, and potential downstream tasks.

broadens its applicability and eliminates the need for interventions used by existing methods, whose cost scales with LLM size. We quantify the efficiency advantage of our regression-based method through a theoretical analysis of its computational cost compared to state-of-the-art intervention-based approaches, establishing conditions under which our method guarantees greater efficiency.

We empirically evaluate `CircuitLasso` against state-of-the-art baselines on a circuit learning benchmark, demonstrating substantial improvements in efficiency while accurately capturing circuits involving LLM components. Leveraging its scalability, we then apply our method to SAE features and obtain human-interpretable circuits for the CoLA dataset, which has not been used before in mechanistic interpretability studies. Finally, we validate the utility of our learned circuits by leveraging them to improve performance in a downstream application of domain generalization.

Our primary contributions are as follows: 1) Inspired by causal graph discovery literature, we introduce an efficient sparse circuit discovery method, `CircuitLasso`, and theoretically analyze its computational cost compared to intervention-intensive approaches. 2) `CircuitLasso` facilitates operating on monosemantic but high-dimensional SAE features, potentially offering clearer explanations of how human-interpretable concepts propagate through LLMs. 3) Extensive experiments across LLMs (up to 9B parameters), SAEs, and datasets demonstrate the efficiency and effectiveness of `CircuitLasso` for circuit discovery and improving generalization performance.

## 2 RELATED WORKS

**Mechanistic Interpretability.** Much of the established work in mechanistic interpretability explains behaviors in terms of raw or coarse-grained model components. Olsson et al. (2022) implicated induction heads in in-context learning, while others (Meng et al., 2022; Geva et al., 2023; Nanda et al., 2023) examined MLP modules for factual recall. However, due to the polysemantic nature of raw neurons and coarse-grained components (Elhage et al., 2022), the resulting mechanistic insights are often difficult to apply to downstream tasks. Some prior methods (Geiger et al., 2023; Zou et al., 2023) attempt to address this issue by fitting model internals to pre-defined hypotheses using curated data, but these approaches fail to generalize to scenarios where researchers lack expert knowledge or cannot anticipate how models implement specific behaviors. Recent work (Bricken et al., 2023; Cunningham et al., 2023) leverages advances in dictionary learning for interpretability and introduces sparse autoencoders (SAEs) to identify sparse, disentangled features in high-dimensional spaces that align with human-interpretable concepts. Building on this, a number of advanced strategies for learning SAE features have been proposed (Rajamanoharan et al., 2024; Gao et al., 2024; Bussmann et al., 2024; Dunefsky et al., 2024). Despite this progress, existing mechanistic interpretability methods continue to face challenges in scaling to the high-dimensional SAE feature space.

**Circuit Learning.** Intervention-based circuit learning approaches, including causal mediation analysis (Vig et al., 2020; Geva et al., 2023; Hanna et al., 2024) and causal tracing (Meng et al., 2022), rely on counterfactual intervention techniques such as activation patching (Nanda, 2023; Syed et al., 2023) to quantify the influence of one component on another. However, these interventions are computationally intensive, making it difficult to scale such methods to large sets of components, let alone to the high-dimensional space of sparse autoencoder (SAE) features. To address this, Marks et al. (2025) propose efficient approximations of intervention-based methods for SAE features, but in high-dimensional settings they must resort to heuristic pre-processing steps such as clustering. These limitations highlight the need for more efficient techniques to learn circuits in LLMs. For example, Laptev et al. (2025a) propose a data-free approach that constructs circuit graphs using information from SAE decoder weights. While many prior works borrow causal concepts (Meng et al., 2022), the broader causal discovery literature has been less explored in the context of circuit learning. Notably, Conmy et al. (2024) iteratively prune edges from the computation graph, reminiscent of constraint-based causal discovery algorithms such as the PC algorithm (Pearl et al., 2000).

## 3 CIRCUIT LEARNING FRAMEWORK AND METHODOLOGY

### 3.1 CIRCUIT DISCOVERY FROM A CAUSAL PERSPECTIVE

In this work, we formulate circuit discovery as the task of learning a directed acyclic graph (DAG) from data, analogous to approaches in causal discovery. State-of-the-art approaches quantify the importance of hidden representations or computational graph edges by estimating their causal effects, particularly indirect effects, using techniques such as causal mediation analysis (Vig et al., 2020), causal tracing (Meng et al., 2022), attribution patching (Nanda, 2023; Syed et al., 2023), and related variants (Kramár et al., 2024; Hanna et al., 2024). These approaches share some similarities with constraint-based causal discovery, which assesses potential edges among variables via independence tests and retains those with strong dependencies. However, constraint-based causal discovery methods are known to face scalability challenges, and circuit discovery methods share this limitation since they must separately quantify the importance of every hidden representation and edge in the computational graph, which can quickly become infeasible with larger models.

Inspired by the continuous causal discovery literature, we propose a (causal) circuit discovery approach, `CircuitLasso`. Assume we extract $N$ components (which may be MLP neurons, attention heads, or SAE features) from all the desired locations in the LLMs and concatenate their activations to form a vector $\boldsymbol{x} = [x_1, x_2, \cdots, x_N] \in \mathbb{R}^N$. Our goal is to learn the DAG $\mathcal{G}$ with the $N$ components as its nodes. We leverage structural equation models (SEM) from continuous causal discovery literature to model the causal relations between a component $x_i$ and its parents $Pa_{\mathcal{G}}(x_i)$: $x_i = f_i(Pa_{\mathcal{G}}(x_i)) + \varepsilon_i$, where $f_i(\cdot)$ is the causal function and $\varepsilon_i$ is the exogeneous noise. In this work, we assume the causal relations between components are linear. Given $M$ observations of the $N$ components, i.e., input matrix $\boldsymbol{X} \in \mathbb{R}^{N \times M}$, we can then obtain the linear SEM in its matrix form:

$$\boldsymbol{X} = A^\top \boldsymbol{X} + \boldsymbol{\varepsilon}, \tag{1}$$

with continuous parameters $A \in \mathbb{R}^{N \times N}$, a weighted adjacency matrix; $\boldsymbol{\varepsilon} \in \mathbb{R}^{N \times M}$ is a matrix of mutually independent exogeneous noises. $A[i, j] \neq 0$ indicates the existence of the causal relation $x_i \to x_j$. We aim to learn $A$ by minimizing the reconstruction error between $\boldsymbol{X}$ and $A^T \boldsymbol{X}$ subject to sparsity and acyclicity constraints:

$$\widehat{A} = \arg\min_A \|\boldsymbol{X} - A^\top \boldsymbol{X}\|_F^2 + \lambda \|A\|_1, \text{subject to } \mathcal{G}(A) \in \mathbb{D} \tag{2}$$

where $\|\cdot\|_F$ denotes Frobenius norm; $\|A\|_1$ is the sparsity penalty with tuning coefficient $\lambda$; $\mathcal{G}(A)$ is the circuit structure inferred from $A$; and $\mathbb{D}$ is the space of acyclic graphs with $N$ nodes. We leverage the established identifiability conditions (Peters et al., 2014) by assuming causal sufficiency and specific noise characteristics (non-Gaussian or equal-variance Gaussian) to ensure the learned DAG is uniquely identifiable and thus interpretable as the underlying causal structure. Thus far, in Eq. (2), we have not made assumptions about the causal ordering of the components in $\boldsymbol{x}$, i.e., the orientations of potential edges between components. Thus, we require the acyclicity constraint $\mathcal{G}(A) \in \mathbb{D}$ to prevent self-loops and cycles, which are unsuitable for interpreting the transmission, aggregation, and evolution of model components. However, the acyclicity constraint is the main computational challenge in solving the circuit learning problem in Eq. (2). In this paper, we make

simplifying assumptions that remove this constraint and reduce the optimization to sparse linear regression problems (also known as Lasso), enabling a scalable solution. We discuss the formulations for circuit discovery on model neurons and SAE features in Sections 4.1 and 3.3.2, respectively.

## 3.2 CIRCUIT DISCOVERY ON NEURONS

To better understand how models encode and process information, mechanistic interpretability research (Conmy et al., 2023a; Cao et al., 2021; Syed et al.) has focused on identifying graphical structures (circuits) connecting pre-trained language model neurons, including outputs from attention and MLP modules. To evaluate the effectiveness of our proposed method in Section 3.1, we follow the same setting as these prior works and treat model neurons as the components of interest. We first collect neuron activations $[\boldsymbol{h}_1, \boldsymbol{h}_2, \ldots, \boldsymbol{h}_L]$ from $L$ target locations, each with dimension $d$, and for $M$ LLM inputs (observations), resulting in $\boldsymbol{H} \in \mathbb{R}^{L \times d \times M}$. Existing circuit discovery methods typically assume that circuit structures respect the model locations' computation order, meaning that neurons from layer $i$ precede those from layer $j$ if $i < j$, and within each layer, attention activations come before MLP activations. We adopt this assumption to simplify the acyclicity constraint in Eq. (2). Accordingly, we reorder $\boldsymbol{H}$ according to the computational graph and reshape it to obtain $\tilde{\boldsymbol{H}} \in \mathbb{R}^{N \times M}$, whereby $N = Ld$. Substitute $\boldsymbol{X}$ in Eq. (2) with $\tilde{\boldsymbol{H}}$ to estimate the weighted adjacency matrix $A$ as

$$\widehat{A} = \arg\min_A \|\tilde{\boldsymbol{H}} - A^\top \tilde{\boldsymbol{H}}\|_F^2 + \lambda \|A\|_1, \quad \text{subject to } A \text{ being block lower triangular.} \tag{3}$$

Specifically, each block $A[i, j]$ is now a $d \times d$ square matrix, and $A[i, j] = 0^{d \times d}$ whenever $i \leq j$. This block lower triangular structure ensures that each block of variables depends only on preceding blocks, so later layers cannot influence earlier ones, thereby preserving the causal ordering without requiring an explicit acyclicity constraint. This constraint exploits the known, human-engineered computational order of the LLM. The underlying architectural insight provides a justifiable acyclic constraint that aligns with the inherent feed-forward nature of the network (activations in later layers are computed after, and depend on, those in earlier layers). Causal discovery with such a justified acyclicity leads to more accurate identification of the underlying causal relationships, as demonstrated by empirical results in Table 1. In practice, we enforce this constraint by initializing the upper-triangular blocks to zero matrices and keeping them fixed throughout optimization. The resulting circuit structure $\mathcal{G}$ is then inferred from $\widehat{A}$.

We now provide complexity analysis of our proposed circuit discovery approach on model neurons versus the existing intervention-based approaches. For our optimization problem in Eq. (3), we have:

**Proposition 3.1.** *Complexity of* `CircuitLasso` *on Model Neurons. Suppose in Eq. (3), $\tilde{\boldsymbol{H}} \in \mathbb{R}^{N \times M}$ contains $M$ samples of $N$-dimensional features, whereby $N = Ld$. Assume a first-order optimization method is used to solve the problem up to convergence error $\epsilon$, then the computational complexity of solving Eq. (3) is $\mathcal{O}(\frac{ML(L-1)d^2}{2\epsilon})$.*

Please refer to the detailed proof in Appendix A.1. We aim to theoretically compare the computational cost of our `CircuitLasso` with the state-of-the-art intervention-based circuit discovery approach. Specifically, we consider the EAP-ig method, which computes IEs for $L$ locations of model components with dimension $d$. EAP-ig relies on the *attribution patching* technique, which applies a linear approximation to IEs, enabling them to be computed in parallel. While the additional cost of estimating IEs from this linear approximation scales linearly with $L$ and considered to have state-of-the-art efficiency, the primary computational burden arises from the two forward passes and one backward pass required for each evaluation of the LLM. Hence, we first provide a computation cost estimation for the EAP-ig with the same learning problem in Eq. (3) and provide conditions when our `CircuitLasso` has guaranteed efficiency over EAP-ig.

**Proposition 3.2.** *Suppose a transformer-based large language model with $S$ blocks has model neuron dimension of $d$. $n_{token}$ is the tokens sequence length, $h$ is the number of attention heads, $f$ being the feedforward expansion factor, then the approximate computational cost of $M$ observations for EAP-ig is roughly $\mathcal{O}\left(16MSn_{token}d^2(2 + f) + 16MSn_{token}^2 d + MLd\right)$, `CircuitLasso` with computational complexity of $\mathcal{O}(\frac{ML(L-1)d^2}{2\epsilon})$ has guaranteed better efficiency compared to EAP-ig*

*if one of the following conditions hold: 1)* $\sqrt{\frac{L(L-1)}{32S(2+f)\epsilon}} < n_{token} \ll d$. *2)* $n_{token} > \sqrt{\frac{L(L-1)d}{32S\epsilon}}$ *and* $n_{token} \gg d$.

Please refer to the detailed proof in Appendix A.2. Intuitively, **Proposition 3.2** provides a guideline for selecting the number of component locations of interest ($L$) and determining which circuit discovery approach is more efficient.

### 3.3 Circuit Discovery on Sparse Features

Although recent research has shown that learned circuits can carry semantically meaningful information and shed light on how information flows through the model, the polysemantic nature of neurons makes them difficult to interpret (Elhage et al., 2022). To address this, more recent work (Marks et al., 2025; Laptev et al., 2025b) explores circuit discovery on sparse SAE features, but the scalability challenge posed by their high dimensionality remains. To address this issue, we extend our proposed formulation to operate on SAE features.

#### 3.3.1 Preliminaries on Sparse Autoencoder (SAE)

Given a model with $d$-dimensional latent space and neuron activations $\boldsymbol{h} \in \mathbb{R}^d$, an SAE can represent $\boldsymbol{h}$ as a linear combination of sparse features $\boldsymbol{z} \in \mathbb{R}^D, D \gg d$:

$$\boldsymbol{z} = \sigma(\boldsymbol{W}_{\text{enc}}\boldsymbol{h} + \boldsymbol{b}_{\text{enc}}), \quad \hat{\boldsymbol{h}} = \boldsymbol{W}_{\text{dec}}\boldsymbol{z} + \boldsymbol{b}_{\text{dec}},$$

where $\boldsymbol{W}_{\text{enc}}, \boldsymbol{b}_{\text{enc}}$ are encoder parameters and $\boldsymbol{W}_{\text{dec}}, \boldsymbol{b}_{\text{dec}}$ are decoder parameters ; $\sigma(\cdot)$ is a nonlinear activation function. The SAEs are usually trained by minimizing the reconstruction error between model activations $\boldsymbol{h}$ and reconstructed activations $\hat{\boldsymbol{h}}$ subject to a sparsity regularizer:

$$\mathcal{L}_{\text{SAE}}(\boldsymbol{h}, \boldsymbol{z}; \boldsymbol{W}_{\text{enc}}, \boldsymbol{b}_{\text{dec}}, \boldsymbol{W}_{\text{dec}}, \boldsymbol{b}_{\text{dec}}) := \|\boldsymbol{h} - \hat{\boldsymbol{h}}\|_2^2 + \alpha\mathcal{L}_{\text{reg}}(\boldsymbol{z}).$$

Recent work on sparse autoencoder (SAE) features explores various methods—ReLU with $L_1$ regularization, thresholding, top-$K$ selection, and Transcoders, to enforce sparsity and improve interpretability. However, our focus is to uncover the causal relations among learned sparse SAE features, rather than developing new SAE training methods. For this study, we employ the following pre-trained SAEs on small (GPT2-small, Pythia-70M), medium (Gemma-2-2b), and large (Gemma-2-9b) LLMs. Please refer to Appendix C.1 for details of employed models and SAEs.

#### 3.3.2 Layer-wise Sparse Feature Circuit Discovery

In this work we assume that causal relations follow the computation order of the underlying model neurons. To be specific, consider two model neurons with dimension of $d$ and their activations at locations $i$ and $j$, denoted by $\boldsymbol{h}_i, \boldsymbol{h}_j \in \mathbb{R}^d$, where computation at $i$ precedes computation at $j$. We obtain the corresponding SAE features $\boldsymbol{z}_i, \boldsymbol{z}_j \in \mathbb{R}^D$ using trained SAEs:

$$\boldsymbol{z}_i = \sigma(\hat{\boldsymbol{W}}_{\text{enc},i}\boldsymbol{h}_i + \boldsymbol{b}_{\text{enc},i}), \quad \boldsymbol{z}_j = \sigma(\hat{\boldsymbol{W}}_{\text{enc},j}\boldsymbol{h}_j + \boldsymbol{b}_{\text{enc},j}).$$

If a causal relationship exists between variables in $\boldsymbol{z}_i$ and $\boldsymbol{z}_j$, we constrain its direction to be from $i$ to $j$. Given $M$ observations of $\boldsymbol{z}_i$ and $\boldsymbol{z}_j$, we obtain input data $\boldsymbol{Z}_i \in \mathbb{R}^{D \times M}$ and $\boldsymbol{Z}_j \in \mathbb{R}^{D \times M}$. We estimate these relations by solving:

$$\widehat{A}_{i,j} = \arg\min_{A_{i,j}} \|\boldsymbol{Z}_j - A_{i,j}^\top \boldsymbol{Z}_i\|_F^2 + \lambda\|A_{i,j}\|_1, A_{i,j} \in \mathbb{R}^{D \times D}. \tag{4}$$

This procedure is repeated for every pair $(i, j)$ where $i$ precedes $j$ in the computation order. In particular, learning $A_{i,j}$ for all transformer block outputs in consecutive layers provides insight into how semantic concepts are transferred, propagated, and evolved across the model. The computational cost of the learning problem in Eq. (4) is $\mathcal{O}(\frac{MD^2}{\epsilon})$.

We also incorporate the downstream prediction target into circuit discovery to enable explanation of the model's predictive behavior. We formulate the following optimization problem to learn a model for predicting the downstream target $y$ using sparse atuoencoder features $\boldsymbol{z}_i$, derived from model neuron activations at location $i$:

$$\widehat{A}_{i,y} = \arg\min_{A_{i,y}} \mathcal{L}_{\text{pred}}(\boldsymbol{y}, A_{i,y}^\top \boldsymbol{Z}_i) + \lambda\|A_{i,y}\|_1, A_{i,y} \in \mathbb{R}^D, \tag{5}$$

where $\mathcal{L}_{\mathrm{pred}}(\cdot,\cdot)$ denotes the prediction loss, instantiated as mean squared error for regression tasks and cross-entropy loss for classification tasks. Application-wise, with the learned $\hat{A}_{i,y}$ and interpretable sparse features $z$, we can not only explain the model's predictive behavior, but also rectify the prediction model to mitigate spurious or biased behavior. The computational cost of the learning problem in Eq. (5) is $\mathcal{O}(\frac{MD}{\epsilon})$.

## 4 EXPERIMENTS

We evaluate `CircuitLasso` on both model neurons (Section 4.1) and sparse autoencoder features (Section 4.2), demonstrating its effectiveness in accurately capturing relations among model components with improved efficiency, providing interpretable insights into model behavior in grammaticality classification, and enhancing downstream performance on a domain generalization task.

### 4.1 CIRCUIT DISCOVERY ON MODEL NEURONS

**Models and Baselines.** We first evaluate our circuit discovery methods on the INTERPBENCH benchmark datasets (Gupta et al., 2024). INTERPBENCH consists of semi-synthetic yet realistic transformers with known circuits, designed for assessing circuit discovery approaches. The transformer models are trained to align their internal computation with a target high-level causal model while constraining non-circuit nodes from influencing the output. We compare our method against four state-of-the-art circuit discovery approaches: Automatic Circuit DisCovery (ACDC) (Conmy et al., 2023b), Subnetwork Probing (SP) (Cao et al., 2021) on nodes and edges, Edge Attribution Patching (EAP) (Syed et al., 2023), and EAP with integrated gradients (EAP-ig) (Marks et al., 2025). While INTERPBENCH includes 86 semi-synthetic transformer models, we follow the protocol of Gupta et al. (2024) and evaluate on the 5 randomly selected cases, which have been empirically verified to be sufficiently realistic for benchmarking circuit discovery techniques.

**`CircuitLasso` on Neurons.** We begin by collecting activations from the model neurons. To ensure fairness, we use the same set of input prompts as the baselines. Given data, we aim to learn a weighted adjacency matrix $A$ that encodes the causal relations between neuron locations, and use it to infer the causal circuit. Please refer to the detailed learning procedure in Appendix B.

**Metrics and Implementation Details.** We evaluate circuit discovery accuracy using the Structural Hamming Distance (SHD) between the ground-truth and estimated circuits, and assess efficiency by measuring runtime in seconds. All experiments were performed over three trials on an NVIDIA A100 machine. We report the mean and standard deviation results in the Table 1.

| Cases | ACDC | | SP | | EAP | | EAP-ig | | CircuitLasso | |
|---|---|---|---|---|---|---|---|---|---|---|
| | SHD | Runtime | SHD | Runtime | SHD | Runtime | SHD | Runtime | SHD | Runtime |
| 3 | $7.6_{\pm 0.06}$ | $78.9_{\pm 2.81}$ | $9.0_{\pm 0.18}$ | $112.5_{\pm 4.89}$ | $12.2_{\pm 0.25}$ | $25.1_{\pm 3.18}$ | $5.2_{\pm 0.13}$ | $42.2_{\pm 4.03}$ | $\mathbf{4.9_{\pm 0.14}}$ | $\mathbf{10.6_{\pm 0.03}}$ |
| 4 | $13.7_{\pm 3.50}$ | $118.8_{\pm 8.45}$ | $15.3_{\pm 3.91}$ | $92.4_{\pm 6.82}$ | $19.8_{\pm 4.76}$ | $58.1_{\pm 7.69}$ | $11.1_{\pm 2.25}$ | $61.9_{\pm 9.16}$ | $9.4_{\pm 2.88}$ | $\mathbf{18.34_{\pm 5.30}}$ |
| 8 | $9.8_{\pm 2.16}$ | $121.5_{\pm 9.81}$ | $11.0_{\pm 1.88}$ | $204.9_{\pm 17.92}$ | $20.9_{\pm 4.92}$ | $54.8_{\pm 2.83}$ | $\mathbf{7.2_{\pm 1.29}}$ | $87.2_{\pm 7.19}$ | $7.4_{\pm 0.96}$ | $\mathbf{50.2_{\pm 6.43}}$ |
| 11 | $5.5_{\pm 0.86}$ | $89.1_{\pm 11.97}$ | $8.7_{\pm 1.47}$ | $117.6_{\pm 9.81}$ | $11.7_{\pm 2.91}$ | $68.7_{\pm 5.08}$ | $4.3_{\pm 0.89}$ | $72.1_{\pm 6.34}$ | $\mathbf{3.9_{\pm 0.22}}$ | $\mathbf{21.3_{\pm 3.08}}$ |
| 101 | $14.2_{\pm 5.11}$ | $90.7_{\pm 7.21}$ | $19.5_{\pm 5.14}$ | $118.5_{\pm 18.16}$ | $20.9_{\pm 4.92}$ | $48.9_{\pm 6.01}$ | $10.9_{\pm 4.58}$ | $51.7_{\pm 5.48}$ | $\mathbf{9.3_{\pm 2.91}}$ | $\mathbf{20.8_{\pm 2.50}}$ |

Table 1: The circuit discovery performance in terms of efficiency (runtime in seconds) and accuracy (SHD) on 5 cases from INTERPBENCH.

According the empirical result in Table 1, we can establish that CMINT is capable of identifying causal relations between model components with accuracy on par with intervention-based approaches, while simultaneously offering markedly improved computational efficiency.

### 4.2 CIRCUIT DISCOVERY ON SPARSE AUTOENCODER FEATURES

We now turn to applying `CircuitLasso` to popular pre-trained LLMs and more realistic tasks, having shown its efficiency advantages over intervention-based methods in Section 4.1. We adapt `CircuitLasso` to learn causal circuits on sparse autoencoder features (as described in Section 3.3.2), revealing model behaviors in terms of human-interpretable concepts (Section 4.2.1). Following Marks et al. (2025), we also leverage insights from learned circuits to improve domain generalization (Section 4.2.2).

### 4.2.1 CASE STUDIES OF PROVIDING INTERPRETATION

**Data and Model.** We demonstrate our approach on the **Co**rpus of **L**inguistic **A**cceptability (CoLA) task (Warstadt et al., 2018) from the GLUE benchmark (Wang et al., 2018), aiming to reveal the inner workings of gpt2-small (Nanda & Bloom, 2022) through interpretable features derived from OpenAI's pre-trained sparse autoencoders for gpt2-small. The CoLA dataset is (to our knowledge) a new dataset for mechanistic interpretation studies. It contains 10,657 sentences from 23 linguistics publications, annotated for grammaticality by the original authors. We conduct our interpretability experiments on the 8,551 training sentences in the public release.

**`CircuitLasso` on Sparse Autoencoder Features.** We extract gpt2-small's neuron activations on the $M$ training sentences and corresponding sparse autoencoder features. Please refer to the detailed learning procedure in Appendix B.

**Sparse Feature Interpretation within Learned Circuits.** The following paragraphs describe our method for interpreting the learned circuit and its constituent sparse features by tracing backward from the prediction target $y$. Starting with $A_{L,y}$, we select features in $\boldsymbol{z}_L \in \mathbb{R}^D$ that are important for predicting $y$. The measure of importance can be either the absolute coefficients $|A_{L,y}|$, or if we wish to focus on a particular prompt, the Hadamard product $\boldsymbol{s} = |A_{L,y}| \odot |\boldsymbol{z}_L| \in \mathbb{R}^D$ between the absolute coefficients and the absolute activations of $\boldsymbol{z}_L$ for this prompt. Suppose the $k^{\text{th}}$ sparse feature $z_{L,k} \in \boldsymbol{z}_L$ is chosen as an important feature. To uncover the semantic concept encoded by $z_{L,k}$, we apply two complementary, cross-validating procedures:

*Multi-prompt approach.* We identify multiple prompts and the tokens with them that strongly activate $z_{L,k}$. By inspecting the collected tokens, we infer the semantic concept encoded by $z_{L,k}$. For example, words ending in "-self" consistently activate the sparse feature $z_{12,20726}$, suggesting that this feature captures the presence of such words. Examples are illustrated in Table 5, 6, and 7 in Appendix C.4.

*Single-prompt approach.* To validate the plausibility of the identified semantic concept for $z_{L,k}$, we select a single prompt, systematically vary one or more of its tokens according to the concept, and observe the resulting changes in $z_{L,k}$. If altering the tokens causes $z_{L,k}$ to lose activation, the inferred concept is considered reasonable. For example, in the prompt "He said that himself was hungry," the word "himself" (the fifth token) activates $z_{12,20726}$ to a value of 1.3509. Replacing "himself" with "him", "he", or other alternatives without the suffix "-self" reduces the activation of $z_{12,20726}$ to 0. An example is illustrated in Table 8 in Appendix C.4.

With the above *multi-prompt* and *single-prompt* approaches, we identify the semantics of important sparse features in the final layer. For example, feature $z_{12,20726}$ captures the concept of "-self", $z_{12,3092}$ corresponds to "ending punctuation", $z_{12,776}$ to "thirst/hunger", and $z_{12,19322}$ to "tired/weary". For each important feature $z_{L,k}$, we then trace its most influential parent variables in the previous layer $L-1$ using the learned adjacency matrix $A_{L-1,L}$. As with layer $L$, we may choose to focus on the current prompt and define the importance measure as the product $\boldsymbol{s} = |\frac{\partial z_{L,k}}{\partial \boldsymbol{z}_{L-1}}| \odot |\boldsymbol{z}_{L-1}| \in \mathbb{R}^D$. We then select the most important features (for example $z_{11,6368}$, $z_{11,29778}$, $z_{11,29041}$, and $z_{11,21518}$), and interpret their semantics using the same *multi-prompt* and *single-prompt* procedures. Repeating this process across all consecutive pairs of layers yields tree-shaped circuit paths spanning the transformer blocks, offering intuition into how semantic concepts are encoded, propagated, and ultimately contribute to task-specific predictions.

Figure 2 presents such a tree-shaped circuit, consisting of sparse features with human-interpretable meanings across 5 layers. More examples are provided in Appendix C.5. From the circuits in Figure 2, we make the following observations:

**Persistence.** Certain semantic concepts persist along circuit paths across multiple layers, particularly in the later layers. For example, the concept of "-self" is present in the 20726$^{\text{th}}$ feature of layer 12, the 6368$^{\text{th}}$ feature of layer 11, the 2985$^{\text{th}}$ feature of layer 10, the 9592$^{\text{th}}$ feature of layer 9, and the 15186$^{\text{th}}$ feature of layer 8. We highlight circuit paths that capture persistence relations between consecutive layer features in black.

**Merging and Dropping.** We also observe that sparse features in later layers can merge semantic concepts from multiple parent features in the preceding layer, or disregard (i.e., drop) certain concepts contributed by those parent features. For instance, the 10609$^{\text{th}}$ feature of layer 9 merges

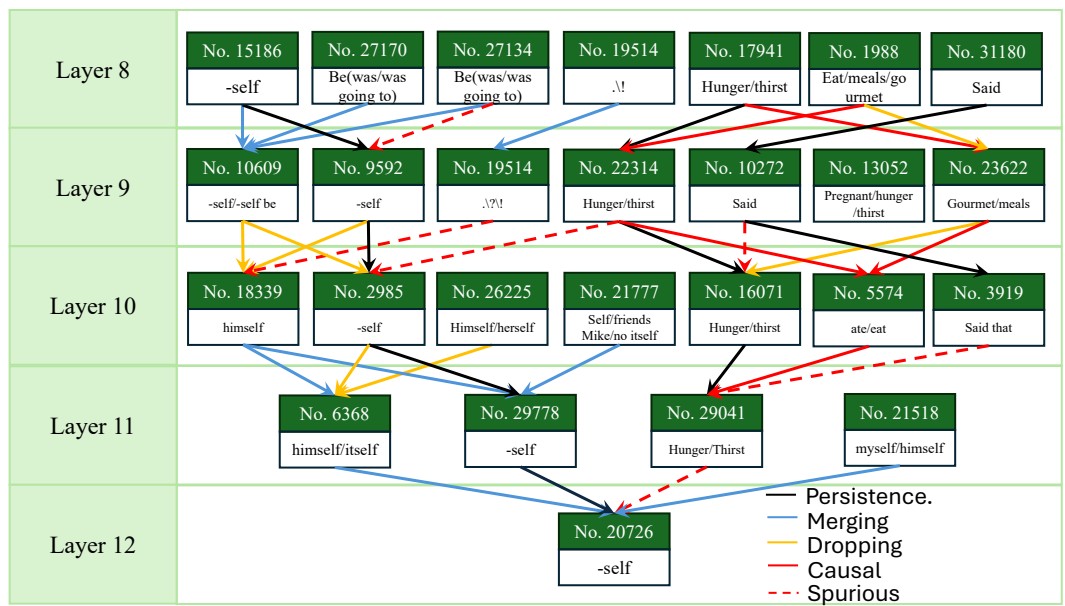

Figure 2: The learned circuits over SAE features on GPT2-small model. Different colors represent different types of edges in the computation graph.

concepts from both the 15186[th] and 27170[th] features of layer 8. In contrast, the 6368[th] feature of layer 11 retains only the concept "himself" and disregards all other forms of "-self" from the 2985[th] feature of layer 10. We highlight circuit paths representing propagation in blue and decomlocation in orange.

**Cause-Effects and Spurious Correlations.** Our circuits can capture causal relations between features that encode cause–effect semantic concepts. For example, the 22314[th] feature of layer 9 represents the concept of "hunger/thirst," which can be considered a cause of the action "ate/eat," encoded in the 5574[th] feature of layer 10. However, our assumption that causal orientatations align with the computation order results in some circuit paths appearing anti-causal. From a human perspective, one typically feels hungry before taking actions such as "eat food/meals/gourmet," yet our circuit includes a path from the 1988[th] feature of layer 8 to the 22314[th] feature of layer 9, which implies the reverse. Moreover, our circuits also capture spurious correlations. For example, the 29041[th] feature of layer 12, which represents "-self," is spuriously correlated with the "hunger/thirst" concept encoded in the 29041[th] feature of layer 11. Such correlations are likely introduced by biases in the training data, such as the frequent co-occurrence of these two semantic concepts within the same sentence. By analyzing these circuit paths, we can infer the nature of dataset biases and potentially mitigate them through targeted model editing. We next show how such insights from a learned circuit can be leveraged to improve downstream domain generalization in Section 4.2.2.

**Faithfulness and Completeness.** To more comprehensively evaluate the quality of our learned circuits on SAE features, we further assess them on the CoLA dataset using the **faithfulness** and **completeness** metrics, following the standard protocol in Marks et al. (2025). In particular, we introduce a new ablation strategy: rather than ablating features outside the circuit by replacing them with their dataset-average values, we ablate edges by removing their direct contributions to the output. Let the learned circuit be $C$, and define the model output $m = p(Y = \text{grammatically correct}) - p(Y = \text{grammatically wrong})$. We first apply the standard feature ablation method of Marks et al. (2025) and compare our results with the intervention-based circuit-learning method SHIFT. To ensure fairness, we exclude SAE reconstruction errors and attention/MLP SAEs from SHIFT. For our `CircuitLasso` approach, we focus only on SAE features within the learned circuit and still ablate features in the original LLM. The top two plots in Figure 3 show the node ablation results. Our learned circuit achieves performance comparable to SHIFT, consistent with the findings in Marks et al. (2025) that relatively small feature circuits can explain a substantial portion of a model's be-

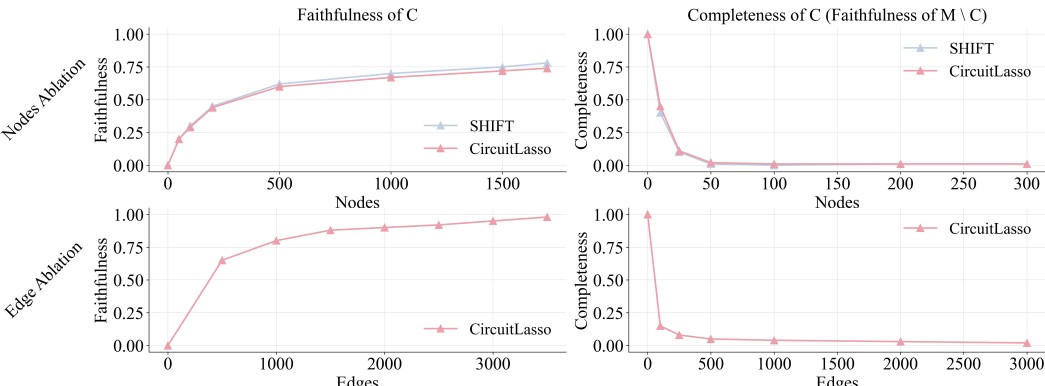

Figure 3: Faithfulness and completeness scores for the learned circuits, evaluated on the CoLA training dataset using both node ablation and novel-edge ablation. Ideal faithfulness is 1, and ideal completeness is 0.

havior. We then perform edge ablation on our circuit. Because our sparse-regression framework explicitly learns edge coefficients, representing the direct influence of each SAE feature on the output, we can ablate a specific edge by setting its coefficient to zero, thereby removing that dependence.[1] The bottom two plots in Figure 3 report these results. The conclusions mirror those from node ablation: a small subset of essential edges, together with their corresponding SAE features, governs the model's prediction behavior.

### 4.2.2 DOWNSTREAM TASK: DOMAIN GENERALIZATION

A common approach in domain generalization is to construct predictors that are robust across domains by removing unintended, domain-variant features, which may be predictive in-distribution but fail to generalize. Inspired the ability of `CircuitLasso` to identify and interpret important features for predicting the downstream target, we investigate whether `CircuitLasso` can remove spurious features.

**Data, Models and Baselines.** We evaluate `CircuitLasso` for domain generalization on the same dataset as Marks et al. (2025), the Bias in Bios dataset (BiB) (De-Arteaga et al., 2019). This dataset consists of professional biographies with the task of classifying an individual's profession from their biography. The BiB dataset encodes a spurious semantic concept, gender, based on which two subsets are constructed: an ambiguous set, where profession and gender are strongly correlated[2], and a balanced set, where profession and gender are independent[3]. The goal is to produce a profession classifier that performs accurately on the balanced set but is trained only on the ambiguous set. While Marks et al. (2025) demonstrate their approach on small to moderate LLMs (Pythia-70M (Biderman et al., 2023) and Gemma-2-2B (Team, 2024)), we extend their evaluation to a larger LLM, Gemma-2-9B (Team, 2024). For all three models, we employ the pre-trained SAEs provided by Lan et al. (2024). Following the evaluation protocol in Marks et al. (2025), we compare against several baselines: a predictor trained on the ambiguous set (ORIGINAL); a predictor trained on the balanced set (ORACLE); concept bottleneck probing (CBP, Yan et al., 2023); and spurious human-interpretable feature trimming (SHIFT, Marks et al., 2025). For existing baseline SHIFT, we adopt the variant that operates on SAE features with manual inspection and evaluation, and exclude the versions trained on neurons or without human inspection due to their consistently inferior performance. We report SHIFT results using both the original linear classifier and a retrained classifier.

---

[1]We do not apply edge ablation to SHIFT, since it does not provide edge-level correlations between SAE features and the output.

[2]For example, all professors are assumed to be male, while nurses are assumed to be female.

[3]The balanced set contains equal numbers of male professors, male nurses, female professors, and female nurses.

**CircuitLasso for Domain Generalization.** We select SAE features from a specific location of a pre-trained LLM, such as the transformer output at layer 22 in Gemma-2-2B. We ablate spurious features by setting their values to zero and *directly feed* the resulting SAE feature values into our trained linear classifier. In addition, similar to SHIFT, we also investigate retraining the linear classifier on the ablated SAE features. Please refer to Appendix B for more details.

| Method | Pythia-70M | | | Gemma-2-2B | | | Gemma-2-9B | | |
|---|---|---|---|---|---|---|---|---|---|
| | Profession | Gender | Worst group | Profession | Gender | Worst group | Profession | Gender | Worst group |
| ORIGINAL | 61.9 | 87.4 | 24.4 | 69.6 | 79.5 | 4.1 | 70.8 | 78.2 | 23.4 |
| CBP | 83.3 | 60.1 | 67.7 | 90.1 | **50.2** | 86.8 | 94.7 | **50.0** | 91.5 |
| ORACLE | 93.0 | 49.4 | 91.4 | **95.1** | 50.2 | 91.7 | 95.7 | **50.0** | 90.5 |
| SHIFT | 88.5 | 54.0 | 76.0 | 72.8 | 51.6 | 43.7 | 77.1 | 52.8 | 67.9 |
| SHIFT-retrain | 93.1 | 52.0 | **89.0** | 94.2 | 52.4 | 92.4 | 96.0 | 51.3 | 90.3 |
| CircuitLasso | 90.5 | **50.1** | 75.8 | 77.5 | 50.7 | 50.5 | 81.5 | 50.3 | 69.8 |
| CircuitLasso-retrain | **94.2** | 50.6 | 88.7 | **95.1** | 52.8 | **92.9** | **96.9** | 50.5 | **91.5** |

Table 2: Prediction accuracy with different LLMs and domain generalization methods.

| Method | Pythia-70M | | Gemma-2-2b | | Gemma-2-9b | |
|---|---|---|---|---|---|---|
| | # of features | Runtime (s) ↓ | # of features | Runtime (s) ↓ | # of features | Runtime (s) ↓ |
| SHIFT | 49 | 257.6 | 65 | 371.2 | 71 | 908.4 |
| SHIFT-retrain | | 356.3 | | 476.8 | | 1056.0 |
| CircuitLasso | 41 | 36.5 | 55 | 47.2 | 59 | 107.4 |
| CircuitLasso-retrained | | 61.9 (17.37%) | | 72.5 (15.20%) | | 125.2 (11.98%) |

Table 3: Runtime and numbers of selected features for SHIFT versus our CircuitLasso method. The runtime does not include manual interpretation time.

**Results.** Tables 2 and 3 present the accuracy and efficiency of our approach compared to baselines. Overall, CircuitLasso consistently achieves competitive or superior results, with clear efficiency advantages that become more pronounced as model size increases. The prediction outcomes in Table 2 demonstrate that CircuitLasso can reliably identify spurious correlations from the learned circuit. Both CircuitLasso and CircuitLasso-retrain slightly outperform SHIFT, which we attribute to our design choice of directly feeding SAE features into a linear classifier. This enables prediction using disentangled semantic concepts, allowing more effective ablation of spurious features and retraining. Efficiency results in Table 3 further underscore the strengths of CircuitLasso, as it requires fewer features and substantially less runtime than SHIFT, particularly for large models. These findings confirm that CircuitLasso not only achieves stronger generalization but also scales more efficiently without compromising interpretability.

## 5 CONCLUSION AND DISCUSSION

In this work, we presented a new circuit learning method, CircuitLasso, based on Lasso regression. Our work offers a novel and effective solution to the challenges of polysemanticity and high dimensionality in LLM circuit learning. By shifting the focus from polysemantic neurons to the monosemantic features extracted by SAEs and applying a scalable sparse regression approach, we are able to discover circuits that are both accurate and interpretable. Our method's ability to handle high-dimensional data, its reliance on observational data, and the sparsity of its learned circuits represent significant advantages over existing baselines. We believe this research offers new insights on how LLMs work and can be impactful for various downstream applications, such as domain generalization.

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

# A  THEORETICAL PROOFS

## A.1  PROOF FOR PROPOSITION 3.1

*Proof.* Solving Eq. (3) is equivalent to solve $N$ LASSO problems. The $n^{\text{th}}$ regression has most $p_n \leq \lfloor \frac{n}{d} \rfloor d$ predictors. Per iteration cost for a single regression is roughly $\mathcal{O}(p_n dM)$. For $N$ regressions, we have $\mathcal{O}\big(Md \sum_{n=1}^{N} p_n\big) = \mathcal{O}(\frac{ML(L-1)d^2}{2})$. For a smooth loss function, first-order methods typically require $\mathcal{O}(\frac{1}{\epsilon})$ iterations to reach an error tolerance $\epsilon$. Hence the total computational cost over all iterations is $\mathcal{O}(\frac{ML(L-1)d^2}{2\epsilon})$. $\qquad\square$

## A.2  PROOF FOR PROPOSITION 3.2

*Proof.* Assume $d$ to be model hidden dimension, $n_{token}$ to be the tokens sequence length, $h$ to be the number of attention heads, so head dimension being $k = \frac{d}{h}$, and $f$ being the feedforward expansion factor, for one transformer block:

- Q/K/V projections: $\approx 6n_{\text{token}}d^2$

- $QK^T$ across all heads: $\approx 2n_{\text{token}}^2 d$.

- Attention-weighted $V$ across heads: $\approx 2n_{\text{token}}^2 d$

- Output projection: $\approx 2n_{\text{token}}d^2$

- Feed-forward network (two linear layers $d \rightarrow fd \rightarrow d$: $\approx 4fn_{\text{token}}d^2$

For the forward pass, the asymptotic complexity is approximately:

$$\mathcal{O}(n_{\text{token}}d^2(8 + 4f) + 4n_{\text{token}}^2 d)$$

For the backward pass, the asymptotic complexity is approximately 2 to 3 times forward passes, we take the lower bound here and approximate it as:

$$\mathcal{O}(n_{\text{token}}d^2(16 + 8f) + 8n_{\text{token}}^2 d)$$

Hence, for 2 forward passes and one backward pass of $M$ observations across $S$ blocks, we have computational cost:

$$\mathcal{O}\Big(16MSn_{token}d^2(2 + f) + 16MSn_{token}^2 d\Big)$$

The linear approximation regarding $L$ locations: $\mathcal{O}(MLd)$, which is usually ignorable compared to the computational cost of forward and backward passes. Hence for EAG-ig, the roughly computational cost is $\mathcal{O}\Big(16MSn_{token}d^2(2 + f) + 16MSn_{token}^2 d + MLd\Big)$.

If $n_{token} \gg d$, then the dominant term is $16MSn_{token}^2 d$, to have efficiency advantage, we have to achieve:

$$16MSn_{token}^2 d > \frac{ML(L-1)d^2}{2\epsilon}$$
$$\implies n_{token}^2 > \frac{L(L-1)d}{32S\epsilon} \qquad (6)$$
$$\implies n_{token} > \sqrt{\frac{L(L-1)d}{32S\epsilon}}$$

Else if $d \gg n_{token}$, then the domain term is $16MSn_{token}d^2(2+f)$, to guarantee efficiency, we must have:

$$16MSn_{token}d^2(2+f) > \frac{ML(L-1)d^2}{2\epsilon}$$
$$\implies n_{token}^2 > \frac{L(L-1)}{32S(2+f)\epsilon} \tag{7}$$
$$\implies n_{token} > \sqrt{\frac{L(L-1)}{32S(2+f)\epsilon}}$$

$\square$

## B  EXPERIMENT DETAILS

**CircuitLasso on Neurons.** We begin by collecting activations from the model neurons. To ensure fairness, we use the same set of input prompts as the baselines. For example, in each case, ACDC employs two sets of data inputs: a clean run with $M$ input prompts, each with $n_{token}$ tokens, i.e., an $M \times n_{token}$ array of tokens $\boldsymbol{T}_{clean}$, and a corrupted run of the same dimensionality, an $M \times n_{token}$ array $\boldsymbol{T}_{corrupted}$. Our approach combines these two runs into a single dataset, an array $\boldsymbol{T} = (\boldsymbol{T}_{clean}, \boldsymbol{T}_{corrupted})$ of $2M \times n_{token}$ tokens, which is then used to generate neuron activations at various locations in the LLM. Given $\boldsymbol{T}$ and a pre-trained model, we obtain neuron activations at a location $i$ with shape $d \times 2M \times n_{token}$ and average over tokens to produce activations $\boldsymbol{H}_i \in \mathbb{R}^{d \times 2M}$. Repeating this process across all $L$ locations and sorting them according to the computation order yields $\tilde{\boldsymbol{H}} \in \mathbb{R}^{Ld \times 2M}$. Substituting $\tilde{\boldsymbol{H}}$, the collected data matrix with $N = Ld$ dimensions and $2M$ observations into Eq. (3), we aim to learn a weighted adjacency matrix $A \in \mathbb{R}^{N \times N}$ that encodes the causal relations between neuron locations. Finally, we infer the causal circuit.

**CircuitLasso on Sparse Autoencoder Features.** We extract gpt2-small's neuron activations on the $M$ training sentences. We select the final outputs from each transformer block (layer) as our locations of interest. Given a prompt with $n_{token}$ tokens, we obtain transformer outputs at the $i^{th}$ layer with shape $d \times n_{token}$ and the corresponding sparse autoencoder features with shape $D \times n_{token}$. We then collect sparse autoencoder features for all $M$ prompts and average across tokens to produce sparse feature activations $\boldsymbol{Z}_i \in \mathbb{R}^{D \times M}$. Repeating this across all $L$ layers yields our dataset $\{\boldsymbol{Z}_1, \boldsymbol{Z}_2, \ldots, \boldsymbol{Z}_L\}$. In our setting, $L = 12$, $d = 768$, $D = 32{,}768$, and $M = 8{,}551$. For the CoLA task, the prediction target $y$ indicates whether a sentence is linguistically acceptable. Substituting $\{\boldsymbol{Z}_i\}_{i=1}^{L}$ and $y$ into Eq. (4) and Eq. (5), we learn weighted adjacency matrices between consecutive layers, $\{A_{i,i+1}\}_{i=1}^{L-1}$, and the weighted adjacency matrix between the final layer sparse autoencoder features $\boldsymbol{z}_L$ and the prediction target $y$, i.e., $A_{L,y}$.

**CircuitLasso for Domain Generalization.** We select SAE features from a specific location of a pre-trained LLM, such as the transformer output at layer 22 in Gemma-2-2B. For each prompt, we average the $D$-dimensional SAE features at this location over tokens and collect them across all $M$ prompts in the training data, producing $\boldsymbol{Z}_s \in \mathbb{R}^{D \times M}$. Substituting $\boldsymbol{Z}_s$ and the target observations $y = (y_1, y_2, \cdots, y_M) \in \mathbb{R}^M$ into Eq. (5), we estimate the weighted adjacency matrix $A_{s,y} \in \mathbb{R}^D$ and identify important features with large absolute coefficients. This process is equivalent to training a sparse linear classifier on the SAE features, which we later exploit for profession prediction. From $A_{s,y}$, we further interpret the semantics of the important features using our proposed *multi-prompt* and *single-prompt* approaches (Section 4.2.1) and manually identify spurious features associated with gender. Unlike the SHIFT method of Marks et al. (2025), which *decodes* SAE features into neuron activations after ablation, we ablate spurious features by setting their values to zero and *directly feed* the resulting SAE feature values into our trained linear classifier. In addition, similar to SHIFT, we also investigate retraining the linear classifier on the ablated SAE features.

# C INTERPRETABILITY ON SPARSE FEATURES CIRCUITS

## C.1 PRELIMINARY OF SPARSE AUTOENCODERS

Cunningham et al. (2023) uses a ReLU activation with $L_1$ sparsity regularization. Subsequent work explores alternative activation functions $\sigma(\cdot)$ to extract desired SAE features. Rajamanoharan et al. (2024) introduces a threshold to determine the minimum pre-activation for feature activation, while Gao et al. (2024) and Bussmann et al. (2024) enforce sparsity by selecting the top $K$ features. Dunefsky et al. (2024) proposes Transcoders, which are similar to SAEs, but focusing on training interpretable approximations of MLPs. In this work, we employ the following pre-trained LLMs and SAEs:

- The open-source GPT2-small SAEs for all sublayers of the open-weights GPT2-small model. These SAEs use a ReLU-linear encoder with $D = 32768$ and $L_1$ sparsity regularization.
- The open-source Pythia-70M SAEs for all sublayers of the open-weight Pythia-70M. These SAEs use a ReLU-linear encoder with $D = 64 \times d$ and $L_1$ sparsity regularization.
- The open-source Gemma Scope SAEs for all sublayers of the open-weights Gemma-2-2B, Gemma-2-9B models. These SAEs use the JumpReLUlinear encoder and set $D = 8 \times d$.

## C.2 ABLATION STUDY OF SPARSITY CONSTRAINT

Table 4 presents an ablation study on the sparsity coefficient $\lambda$ for circuit discovery between the last layer sparse features and prediction target. When $\lambda = 0$, the model achieves the highest training accuracy (99.36%) but suffers severe overfitting, as reflected in a large performance drop on the test set (71.10%). Introducing a small sparsity constraint ($\lambda = 10^{-5}$) improves test accuracy to 72.77%, the best among all settings, indicating enhanced generalization. Larger values of $\lambda$ further enforce sparsity but lead to higher training loss and a notable decline in both training and test accuracy, suggesting that excessive sparsity harms the model's capacity to capture meaningful circuit structure. We therefore select $\lambda = 10^{-5}$ as the optimal setting, as it achieves the best test accuracy and generalization ability while preserving a sparse, interpretable circuit structure.

| $\lambda$ | Training Set | | | Test Set | |
|---|---|---|---|---|---|
| | Prediction Loss | L1 Loss | Prediction Accuracy (%) | Prediction Loss | Prediction Accuracy (%) |
| 0 | 0.0612 | - | **99.36** | 0.9977 | 71.10 |
| $10^{-5}$ | 0.1741 | 9257.6 | 95.70 | 0.6684 | **72.77** |
| $5 \times 10^{-5}$ | 0.3498 | 1898.7 | 85.44 | 0.5566 | 72.48 |
| $5 \times 10^{-4}$ | 0.5535 | 45.3 | 72.14 | 0.5566 | 70.28 |
| $10^{-4}$ | 0.6143 | 0.5 | 70.44 | 0.6283 | 69.14 |

Table 4: Ablation study of sparsity constraint coefficient $\lambda$ for circuit discovery between last layer $z_L$ and prediction target $y$.

## C.3 STATISTICS OF LEARNED CIRCUIT WEIGHTED ADJACENCY MATRICES

Figure 5 examines the coefficient distribution $|A_{L,y}|$ under the best setting ($\lambda = 10^{-5}$). We observe that most coefficient values are extremely small, with 82.2% below 0.01, 83.3% below 0.10, 86% below 0.41, suggesting that only a small subset of features contribute substantially to the prediction. The top 5 essential features clearly dominate the distribution, highlighting the effectiveness of the sparsity constraint in filtering out irrelevant features and isolating semantically interpretable ones.

## C.4 SEMANTIC CONCEPTS ENCODED IN SPARSE FEATURES

## C.5 SPARSE FEATURES CIRCUITS

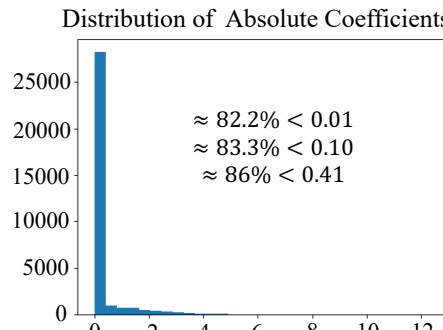

Top 5 Sparse Features Indices

| Indices | Absolute Coefficients |
|---------|----------------------|
| 20726 | 12.3138 |
| 3908 | 10.8914 |
| 3261 | 10.6946 |
| 4628 | 10.5880 |
| 1168 | 10.4870 |

Figure 4: The distribution of $|A_{L,y}|$ and the selected top 5 essential features.

| Prompts | Values of $z_{12,20726}$ | Values of $z_{12,20726}$ for each token |
|---------|------------------------|----------------------------------------|
| Kiss himself . | 0.3051 | (0, 0, 0, 1.5254 , 0) |
| This movie just watches itself . | 0.1861 | (0, 0, 0, 0, 0, 1.3027 , 0) |
| This window just opens itself . | 0.1759 | (0, 0, 0, 0, 0, 1.2311 , 0) |
| This list includes my name on itself . | 0.1697 | (0, 0, 0, 0, 0, 0, 0, 1.5277 , 0) |
| This silver polishes itself . | 0.1692 | (0, 0, 0, 0, 0, 1.1844 , 0) |
| He said that himself was hungry | 0.1689 | (0, 0, 0, 0, 1.3509 , 0, 0, 0) |
| Every picture of itself arrived. | 0.1665 | (0, 0, 0, 0, 1.1652 , 0, 0) |
| Bill understands Mary and himself . | 0.1638 | (0, 0, 0, 0, 0, 1.1467 , 0) |
| Myself saw me | 0.1602 | (0, 0, 0.8009 , 0, 0) |

Table 5: *Multi-prompt* approach for identifying semantic concepts for sparse features $z_{12,20726}$.

| Prompts | Values of $z_{12,776}$ for each token |
|---------|--------------------------------------|
| Hun ger fainted Sharon. | (0, 0.9980 , 3.3986 , 0, 0, 0, 0) |
| Many people were dying of thirst . | (0, 0, 0, 0, 0, 0, 2.0910 , 0) |
| One people was dying of thirst . | (0, 0, 0, 0, 0, 0, 1.8140 , 0) |
| John whined that he was hungry . | (0, 0, 0, 0, 0, 0, 0, 1.9004 , 0) |
| Many soldiers have claimed bottled water satisfies thirst best. | (0, 0, 0, 0, 0, 0, 0, 0, 1.9243 , 0, 0) |

Table 6: *Multi-prompt* approach for identifying semantic concepts for sparse features $z_{12,776}$.

| Prompts | Values of $z_{12,19322}$ for each token |
|---------|----------------------------------------|
| The teacher became tired of the students. | (0, 0, 0, 0, 2.7252 , 1.0383 , 0, 0, 0) |
| The president looked weary . | (0, 0, 0, 0, 2.1888 , 0) |
| Genie intoned that she was tired . | (0, 0, 0, 0, 0, 0, 0, 0, 2.6658 ) |
| John placed him busy . | (0, 0, 0, 0, 1.5510 , 0) |
| Visiting relatives can be boring . | (0, 0, 0, 0, 0, 0, 1.8287 , 0) |

Table 7: *Multi-prompt* approach for identifying semantic concepts for sparse features $z_{12,19322}$.

| Prompts | Values of $z_{12,3092}$ for each token |
|---|---|
| He said that himself was hungry . | (0, 0, 0, 0, 0, 0, 0, 6.0688 ) |
| He said that himself was hungry yet | (0, 0, 0, 0, 0, 0, 0, 0 ) |
| He said that himself was hungry , | (0, 0, 0, 0, 0, 0, 0, 0 ) |
| He said that himself was hungry ? | (0, 0, 0, 0, 0, 0, 0, 3.4722 ) |
| He said that himself was hungry ! | (0, 0, 0, 0, 0, 0, 0, 3.7391 ) |

Table 8: *Single-prompt* approach for identifying semantic concepts for sparse features $z_{12,3092}$.

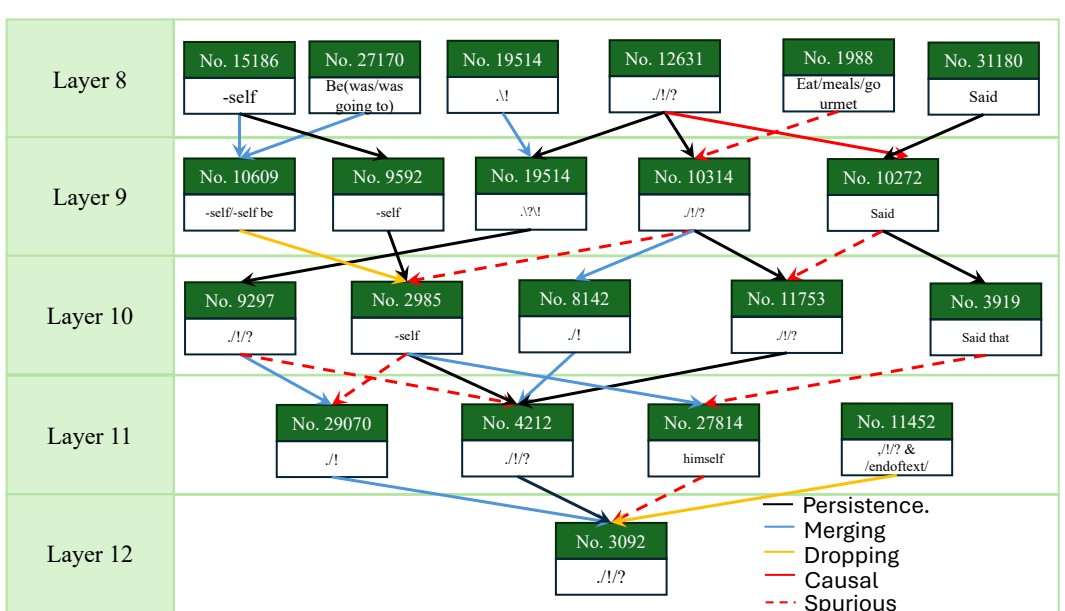

Figure 5: The learned circuit over SAE features on GPT2-small model. Starting with feature No.3072 in layer 12.

