# OpenReview forum: "Scalable Circuit Learning for Interpreting Large Language Models"
_ICLR.cc/2026/Conference — Submitted to ICLR 2026_

### Official Review · Reviewer_jgHf · 2025-10-30

**Soundness:** 2
**Presentation:** 3
**Contribution:** 3
**Rating:** 4
**Confidence:** 2

**Summary:**

The paper proposes CircuitLasso, a scalable circuit-learning method that reframes circuit discovery in LLMs as sparse linear regression over internal components (i.e., SAE features). By assuming a causal ordering aligned with the forward pass and linear relations, the method removes explicit acyclicity constraints and yields a block-lower-triangular Lasso objective. What's more, the authors provide a complexity analysis contrasting CircuitLasso with intervention-based methods, show accuracy/efficiency tradeoffs, then scale to SAE features to analyze GPT-2 small on CoLA, revealing interpretable multi-layer feature flow. Finally, they leverage learned circuits to ablate SAE features and improve domain generalization on Bias-in-Bios across Pythia-70M, Gemma-2-2B, and Gemma-2-9B, with favorable accuracy/runtime w.r.t. baselines.

**Strengths:**

+ It's interesting to see CircuitLasso to cast circuit discovery as a block-structured Lasso aligned with compute order, which avoids explicit acyclicity constraints and costly intervention loops.

+ From the experiments, it seems that CircuitLasso can reveal clear concepts (e.g., “-self”, punctuation, hunger/thirst) and their persistence/merging/dropping across layers, moving beyond polysemantic neuron noise (GPT-2/CoLA).

**Weaknesses:**

- My primary concern is the paper’s reliance on causal discovery without explicitly stating or validating its underlying assumptions. By using a Lasso-based approach while effectively overlooking key conditions, e.g., unconfoundedness and correct causal ordering, the method risks learning spurious or anti-causal circuits rather than the true causal structure. Without targeted interventional tests or invariance checks across environments, such errors may go undetected.

- In addition, a purely linear formulation may miss important nonlinear interactions among features. Therefore, the absence of residual diagnostics or small-scale nonlinear ablation study weakens the method.

**Questions:**

Please refer to my summary of weaknesses.

---

> ### Author Response · Authors · 2025-11-25
> **Responses to reviewer jgHf**
>
> We sincerely appreciate the reviewers’ thoughtful and constructive feedback. We have carefully addressed all concerns with additional analyses, experiments, and clarifications.
>
> **Underlying Assumptions to Guarantee Causality.** Our CircuitLasso approach is grounded in principles of SEM-based causal discovery, hence inevitably inherit assumptions that are generally assumed in SEM-based causal discovery. We leverage the established identifiability conditions (Peters et al., 2014) [1] by assuming causal sufficiency and specific noise characteristics (non-Gaussian or equal-variance Gaussian) to ensure the learned DAG is uniquely identifiable and thus interpretable as the underlying causal structure. We have included these discussions in the paper.
>
> [1] Peters, Jonas, et al. "Causal discovery with continuous additive noise models." The Journal of Machine Learning Research 15.1 (2014): 2009-2053.
>
> **Validity of the Computational Order Constraint.** In SEM-based causal discovery, the matrix $A$ in Eq. (1) models relations among variables, and imposing an acyclic constraint on $A$ yields a directed acyclic graph learned from observational data. In our framework, we replace this general acyclicity constraint with a computational order constraint. We agree that the asymptotic correctness of this imposed order relative to the true (but unknown) causal ordering is important for accurately capturing causal dependencies. However, we would like to clarify that **the computational order we adopt is a reasonable and practically motivated constraint when the true acyclicity structure is unknown.** In particular, our method exploits the known, human-engineered computational order of the LLM. This architectural insight provides a justifiable acyclic constraint that aligns with the inherent feed-forward nature of the network (activations in later layers are computed after, and depend on, those in earlier layers).  Causal discovery with such a justified acyclicity leads to more accurate identification of the underlying causal relationships. On benchmark datasets Interpbench with known ground truth circuits, our approach achieves state-of-the-art graph learning accuracy and outperforms the majority of the existing circuit learning methods. While the optimal acyclic constraint is less clear for SAE features, as their presumed monosemantic nature may conflict with the architectural computation order, our method still provides valuable mechanistic insights.  For instance, Figure 2 illustrates how a specific semantic concept propagates through the model and ultimately influences the target prediction. Even when the ordering is not optimal, we are still able to recover circuit structures that reveal plausible cause-effect pathways and provide useful intuition about the inner workings of LLMs.
>
> **Further Tests and Checks.** In particular, for SAE features, whereby the computational order may not perfectly align with human-interpretable causal relations, While the optimal acyclic constraint is less clear for SAE features, as their presumed monosemantic nature may conflict with the architectural computation order, our method still provides valuable mechanistic insights.  For instance, Figure 2 illustrates how a specific semantic concept propagates through the model and ultimately influences the target prediction. In addition, we further analyze the semantic meaning of the features and validate causal dependencies using additional procedures, namely the **single-prompt** (lines 350-355) approach and **multi-prompt** (lines 345-348) approach. The single-prompt analysis can be viewed as a form of perturbation and intervention based verification. Using these approaches, we can systematically detect and refine the circuit to eliminate anti-causal or spurious correlations (e.g., `ate/eat` leading to `hunger`; `said that` leading to `hunger`), thereby validating and improving the underlying sparse linear model. Moreover, while this work does not explicitly incorporate additional robustness checks for the identified dependencies, our framework is grounded in continuous sparse linear regression. This foundation naturally enables systematic and scalable invariance verification across environments through the adoption of stable invariant learning techniques [2,3].
>
> [2] Ahuja, Kartik, Karthikeyan Shanmugam, and Amit Dhurandhar. "Linear regression games: Convergence guarantees to approximate out-of-distribution solutions." International Conference on Artificial Intelligence and Statistics. PMLR, 2021.
>
> [3] Fan, Jianqing, et al. "Environment invariant linear least squares." The Annals of Statistics 52.5 (2024): 2268-2292.

---

> ### Author Response · Authors · 2025-11-25
>
> **Linearity vs Nonlinearity.** We would like to clarify that our linear formulation can be naturally extended to model nonlinear dependencies by replacing the causal functions $f_i, i=1,2,\cdots, N$ with nonlinear functions. For example, one may adopt a kernel function (kernelized regression) or a multi-layer perceptron, such as the sparse neural DAGs used in Zheng et al. (2020) [4]. However, introducing nonlinearity increases both training time and computational cost because of the larger number of parameters involved. Since our primary interest is the structure of the learned circuit, specifically the connectivity among features rather than precise weight values or fine-grained statistical effects, the linear formulation is sufficient and preferable due to significantly improved efficiency. To illustrate this point, we present a nonlinear extension of our CircuitLasso approach by replacing the linear causal function with a two-layer perceptron:
> $$X_i = W_i^{(2)}\text{ReLU}(W_i^{(1)} X), \quad i = 1,2,\dots,N,$$
> where $X_i \in \mathbb{R}^{1 \times M}$, $X \in \mathbb{R}^{N \times M}$, $W_i^{(1)} \in \mathbb{R}^{m_1 \times N}$, and $W_i^{(2)} \in \mathbb{R}^{1 \times m_1}$. We impose the computational order constraint on the first-layer weights $(W_1^{(1)}, W_2^{(1)}, \dots, W_N^{(1)})$ by enforcing $W_i^{(1)}[:, 1:i-1] = 0.$ We evaluate this nonlinear variant on the InterpBench dataset and compare it with the linear formulation. The results are summarized in the table below. As shown the table below, the nonlinear model learns circuits with similar connectivity patterns and therefore achieves comparable SHD values, supporting our claim that the linear formulation is sufficient for recovering the structural dependencies of interest.
>
> [4] Zheng, Xun, et al. "Learning sparse nonparametric dags." International conference on artificial intelligence and statistics. Pmlr, 2020.
>
> | Case |# Ground-truth Links| EAP SHD | EAP Runtime | EAP-ig SHD | EAP-ig Runtime | CircuitLasso-linear SHD | CircuitLasso-linear Runtime | CircuitLasso-nonlinear SHD | CircuitLasso-nonlinear Runtime |
> |------|------------|------------|---------------|------------|----------------|---------------------------|------------------------------|------------------------------|----------------------------------|
> | 13   | 22 |  12.3  |  37  |  9.3  |  42  |  6.7  |  **14**  |  **6.3**  |  77   |
> | 18   | 2 | 20.0  |  38  |  15.3  |  58  |  **11.0**   |  **20**  |  11.3  |  78  |
> | 19   | 10 |  11.3  |  32  |  9.3  |  40   |  8.0  |  **15**  |  **7.7**   |  83   |
> | 20   | 1 |  **2.0**  |  25  |  **2.0**  |  40  | **2.0** |  **15**  | **2.0** | 67 |
> | 21   | 17 |  23.0  |  39  | 21.7 | 57 | **11.0** | **13** | **11.0**  | 115 |
> | 26   | 2 |  5.3  |  32  | **3.7** | 40 | **3.7** | **20** | **3.7** | 76 |
> | 29   | 1 |  **2.0**  |  22  | **2.0** | 33 | **2.0** | **11** |  **2.0** | 52 |
> | 33   | 1 |  2.7  |  25  | **2.0** | 32 | 2.7 | **15** |  **2.0** | 55 |
> | 34   | 1 |  **2.0**  |  25  | **2.0** | 32 | **2.0** | **11** |  **2.0** | 55 |
> | 35   | 1 |  2.3  |  25  | **2.0** | 32 | 2.3 | **11** |  **2.0** | 53 |
> | 36   | 1 |  2.3  |  24  | **2.0** | 33 | **2.0** | **13** |  **2.0** | 53 |
> | 37   | 1 |  **2.0**  |  25  | **2.0** | 32 | **2.0** | **11** |  **2.0** | 50 |

---

### Official Review · Reviewer_mu8u · 2025-10-30

**Soundness:** 3
**Presentation:** 3
**Contribution:** 3
**Rating:** 6
**Confidence:** 3

**Summary:**

The paper considers the problem of building circuits for interpreting the behavior of Large Language Model. The goal here is to construct a graph denoting causal dependencies between the various internal components of the model which faithfully represent its behavior. Unfortunately, while prior work devised efficient methods to do this for the neurons in a generic LLM, these circuits are hard to interpret since the neurons in a network are often polysemantic and may represent multiple distinct concepts. In order to disentangle these concepts, other interpretability research focuses on interpreting the behavior of individual neurons as sparse combinations of (hopefully) monosemantic concepts (sparse autoencoder or SAE features). However, these techniques are not compatible with prior techniques for circuit construction. The main contribution of the paper is a method to build circuits directly in the feature space of these monosemantic concepts rather than on the polysemantic neural representations.

The main technique in the paper is quite simple. The core idea of the approach is to solve a sparse linear regression problem between the SAE features of successive layers of the model. For each layer of the model, the approach trains a sparse linear regression model which predicts the SAE features at that layer as a linear function of those in the previous layer. By tracing backwards from the output of the model through the features with the largest influence in the previous layer (measured by the coefficients of the sparse regression vector and optionally, with the particular input), the method produces a circuit over the SAE features. The authors also experimentally validate the method in a range of setting, obtaining considerable efficiency gains.

Overall, the approach proposed is natural and efficient. My main concern with the paper is the lack of positional awareness in the learned circuits. While averaging over the representations of individual neurons may enable scalability, this potentially comes at the cost of understanding the causal structure between different parts of the input. This makes a comparison to other interpretability methods, constructing position aware dependency graphs on neuron activations, challenging. I request the authors to expand more on the tradeoffs incurred by this choice. Despite this, I believe this is a good paper and would make a nice addition to the conference.

**Strengths:**

See main review

**Weaknesses:**

See main review

**Questions:**

See main review

---

> ### Author Response · Authors · 2025-11-25
> **Responses to reviewer mu8u**
>
> We sincerely appreciate your careful review of our work and your insightful and encouraging comments.
>
> Regarding your comment on lacking positional awareness in the learned circuits, we would like to clarify that the circuit our approach renders, which focuses on the structure among the consecutive-layer SAE features, can be viewed as a supergraph that can later be specialized to incorporate token-wise structure (positional awareness). In fact, in our interpretability analysis, we take an initial step in this direction by examining how different parts of the input contribute to the activation of specific SAE features, as shown in Tables 5, 6, and 7 in the Appendix. We agree that integrating explicit token-wise modeling is a promising next step. Most importantly, our proposed method, which introduces sparse linear regression to circuit learning, provides the ideal framework for adding positional awareness due to its inherent efficiency and the ease of enforcing structural constraints.

---

### Official Review · Reviewer_gmiW · 2025-10-31

**Soundness:** 3
**Presentation:** 3
**Contribution:** 2
**Rating:** 4
**Confidence:** 4

**Summary:**

The paper introduces CircuitLasso, a scalable method for circuit learning in large language models (LLMs), which leverages sparse linear regression (Lasso) to uncover and interpret relationships among sparse autoencoder (SAE) features. The method addresses challenges of interpretability and scalability posed by high-dimensional, monosemantic SAE features, and avoids computationally expensive interventions used in prior causal circuit discovery methods. CircuitLasso is empirically evaluated on INTERPBENCH and real-world LLMs, demonstrating efficiency and interpretability advantages, and yielding semantically meaningful circuits as well as measurable gains in downstream tasks such as domain generalization.

**Strengths:**

- The paper is generally well-structured, with a clear exposition of the evaluation criteria, algorithmic steps, and experimental setups. It is clear and novel that CircuitLasso reframes circuit discovery as a problem of sparse structure estimation, eliminating reliance on computationally expensive interventions. This makes large-scale analysis of SAE features feasible and efficient, especially as model size increases.

- The method operates directly on SAE latent features, yielding human-interpretable circuits. The CoLA case study presents intuitive, reproducible feature interpretations accompanied by a multi-prompt or single-prompt methodology that readers can replicate.

- The framework is evaluated on both synthetic benchmarks with ground-truth circuits and real-world LLM tasks, confirming its soundness, reliability, and general applicability.

- The learned circuits are shown to be functionally useful, enabling downstream applications like bias mitigation and domain generalization. Removing spurious SAE features leads to improved robustness, demonstrating tangible benefits beyond interpretability.

- The work builds on publicly available SAEs such as GPT-2 and Gemma-Scope, ensuring reproducibility, transparency, and ease of adoption for the broader interpretability community.

**Weaknesses:**

- CircuitLasso relies on a linear structural equation assumption (Section 3.1) and constrains edge directions to follow the model’s forward computation order. While these simplifications enable scalability and convex optimization, they limit causal faithfulness in highly nonlinear or feedback dominated regimes. The method may produce anti-causal or oversimplified paths, for example, hunger→eat vs. eat→hunger in Figure 2.

- Because the framework operates purely on observational activations, it cannot fully distinguish causal relationships from correlated or confounded dependencies. This is particularly concerning for LLMs trained on large, biased, or polysemous corpora, where co-occurrence artifacts are common. Without interventional or perturbation-based validation, the discovered circuits may reflect statistical associations rather than true causal mechanisms.

- Experiments use the five main models of the INTERPBENCH. However, the dataset has 86 different models containing 85 synthetic ones and a non-synthetic one. While the choice is justified, a stronger empirical case would include a subset diversity study and at least one non-synthetic diagnostic beyond CoLA for mechanistic tasks to establish faithfulness.

- The identification of important SAE features via coefficient magnitude or Hadamard-weighted edges is heuristic and may be unstable under feature collinearity or overlapping semantics.

- Furthermore, CircuitLasso inherits limitations of the underlying SAEs. If SAE features are not genuinely monosemantic or exhibit high reconstruction error, the resulting circuits could be misleading. The paper does not quantify how SAE fidelity affects circuit accuracy or interpretability.

**Questions:**

- The paper reports that direct SAE-feature ablation outperforms SHIFT’s decode-then-ablate route on larger models. Could the authors elaborate on the underlying reason, whether it could be reduced reconstruction noise, stronger feature disentanglement, or layer-specific sparsity?

- Could CircuitLasso be extended to capture nonlinear dependencies, for instance through group Lasso, kernelized regressions, or sparse neural DAGs? Where does the runtime advantage come from? Is it primarily a result of sparsity, or could algorithmic choices also apply to nonlinear variants?

- For anti-causal or ambiguous edges, could targeted activation patching or minimal interventional tests help refine causal direction without losing scalability?

- How robust are the learned circuits to the choice of SAE dictionary? Does retraining or altering the SAE dictionary significantly change the inferred circuit topology or key dependencies?

- In addition, an important question concerns the practical scalability of the proposed framework given the high computational cost of training sparse autoencoders. CircuitLasso requires substantial data, compute, and hyperparameter tuning, which may limit the framework’s accessibility and generalizability to other models or domains. Could the authors comment on how the overall pipeline might be made more practical in multiple LLMs?

- Experiments use a subset of 5 cases of the INTERPBENCH models for the main evaluation. However, the paper does not publicly specify the criteria used to select those specific cases from the full set of 86 (or the 16 main ones mentioned). Could the authors clarify how the subset was chosen and whether the selection introduces any bias in comparing circuit-discovery methods?

---

> ### Author Response · Authors · 2025-11-25
> **Responses to reviewer gmiW**
>
> To address the reviewer's concerns, we first provide a high-level clarification covering three key areas: (1) the assumptions required to guarantee causality (referencing the first and second bullet points); (2) the validity of our computational order constraint (first bullet point); and (3) interventional verification and stability/invariance checks (second and fourth bullet points). Specific responses regarding the Interpbench experiments, nonlinear extensions and the SAE are detailed in the responses to the specific questions.
>
> **Underlying Assumptions to Guarantee Causality.** Our CircuitLasso approach is grounded in principles of SEM-based causal discovery, hence inevitably inherit assumptions that are generally assumed in SEM-based causal discovery. We leverage the established identifiability conditions (Peters et al., 2014) [1] by assuming causal sufficiency and specific noise characteristics (non-Gaussian or equal-variance Gaussian) to ensure the learned DAG is uniquely identifiable and thus interpretable as the underlying causal structure. We have included these discussions in the paper.
>
> [1] Peters, Jonas, et al. "Causal discovery with continuous additive noise models." The Journal of Machine Learning Research 15.1 (2014): 2009-2053.
>
> **Validity of the Computational Order Constraint.** In SEM-based causal discovery, the matrix $A$ in Eq. (1) models relations among variables, and imposing an acyclic constraint on $A$ yields a directed acyclic graph learned from observational data. In our framework, we replace this general acyclicity constraint with a computational order constraint. We agree that the asymptotic correctness of this imposed order relative to the true (but unknown) causal ordering is important for accurately capturing causal dependencies. However, we would like to clarify that **the computational order we adopt is a reasonable and practically motivated constraint when the true acyclicity structure is unknown**. In particular, our method exploits the known, human-engineered computational order of the LLM. This architectural insight provides a justifiable acyclic constraint that aligns with the inherent feed-forward nature of the network (activations in later layers are computed after, and depend on, those in earlier layers).  Causal discovery with such a justified acyclicity leads to more accurate identification of the underlying causal relationships. On benchmark datasets Interpbench with known ground truth circuits, our approach achieves state-of-the-art graph learning accuracy and outperforms the majority of the existing circuit learning methods. While the optimal acyclic constraint is less clear for SAE features, as their presumed monosemantic nature may conflict with the architectural computation order, our method still provides valuable mechanistic insights.  For instance, Figure 2 illustrates how a specific semantic concept propagates through the model and ultimately influences the target prediction. Even when the ordering is not optimal, we are still able to recover circuit structures that reveal plausible cause-effect pathways and provide useful intuition about the inner workings of LLMs.
>
> **Further Tests and Checks.** In particular, for SAE features, whereby the computational order may not perfectly align with human-interpretable causal relations, While the optimal acyclic constraint is less clear for SAE features, as their presumed monosemantic nature may conflict with the architectural computation order, our method still provides valuable mechanistic insights.  For instance, Figure 2 illustrates how a specific semantic concept propagates through the model and ultimately influences the target prediction. In addition, we further analyze the semantic meaning of the features and validate causal dependencies using additional procedures, namely the **single-prompt** (lines 350-355) approach and **multi-prompt** (lines 345-348) approach. The single-prompt analysis can be viewed as a form of perturbation and intervention based verification. Using these approaches, we can systematically detect and refine the circuit to eliminate anti-causal or spurious correlations (e.g., `ate/eat` leading to `hunger`), thereby validating and improving the underlying sparse linear model. Moreover, while this work does not explicitly incorporate additional robustness checks for the identified dependencies, our framework is grounded in continuous sparse linear regression. This foundation naturally enables systematic and scalable invariance verification across environments through the adoption of stable invariant learning techniques [2,3].
>
> [2] Ahuja, Kartik, Karthikeyan Shanmugam, and Amit Dhurandhar. "Linear regression games: Convergence guarantees to approximate out-of-distribution solutions." International Conference on Artificial Intelligence and Statistics. PMLR, 2021.
>
> [3] Fan, Jianqing, et al. "Environment invariant linear least squares." The Annals of Statistics 52.5 (2024): 2268-2292.

---

> ### Author Response · Authors · 2025-11-25
>
> > Question 1: The paper reports that direct SAE-feature ablation outperforms SHIFT’s decode-then-ablate route on larger models. Could the authors elaborate on the underlying reason, whether it could be reduced reconstruction noise, stronger feature disentanglement, or layer-specific sparsity?
>
> The key difference between SHIFT and our framework lies in the design of the prediction pipeline. After performing feature ablation, SHIFT reconstructs the original model neurons from the SAE features and makes predictions using these reconstructed neurons. In contrast, our approach constructs a predictor directly on top of the disentangled SAE features, bypassing the neuron reconstruction step entirely. SHIFT’s reconstruction step can obscure the influence of individual feature ablations because model neurons are inherently polysemantic: a single neuron mixes multiple semantic directions. As a result, the weights in a neuron-level predictor do not reliably reflect the isolated contribution of specific SAE features to the target. By operating directly in the SAE feature space, our method avoids this polysemantic entanglement. The learned predictor weights correspond more transparently to dependencies between disentangled concepts and the target, enabling more nuanced and fine-grained ablation analysis.
>
> > Question 2: Could CircuitLasso be extended to capture nonlinear dependencies, for instance through group Lasso, kernelized regressions, or sparse neural DAGs? Where does the runtime advantage come from? Is it primarily a result of sparsity, or could algorithmic choices also apply to nonlinear variants?
>
> We would like to clarify that our linear formulation can be naturally extended to model nonlinear dependencies by replacing the causal functions $f_i, i=1,2,\cdots, N$ with nonlinear functions. For example, one may adopt a kernel function (kernelized regression) or a multi-layer perceptron, such as the sparse neural DAGs used in Zheng et al. (2020) [4]. However, introducing nonlinearity increases both training time and computational cost because of the larger number of parameters involved. Since our primary interest is the structure of the learned circuit, specifically the connectivity among features rather than precise weight values or fine-grained statistical effects, the linear formulation is sufficient and preferable due to significantly improved efficiency. To illustrate this point, we present a nonlinear extension of our CircuitLasso approach by replacing the linear causal function with a two-layer perceptron:
> $$X_i = W_i^{(2)} \text{ReLU}(W_i^{(1)} X), \quad i = 1,2,\dots,N,$$
> where $X_i \in \mathbb{R}^{1 \times M}$, $X \in \mathbb{R}^{N \times M}$, $W_i^{(1)} \in \mathbb{R}^{m_1 \times N}$, and $W_i^{(2)} \in \mathbb{R}^{1 \times m_1}$. We impose the computational order constraint on the first-layer weights $(W_1^{(1)}, W_2^{(1)}, \dots, W_N^{(1)})$ by enforcing $W_i^{(1)}[:, 1:i-1] = 0.$ We evaluate this nonlinear variant on the InterpBench dataset and compare it with the linear formulation. The results are summarized in the table below. As shown in the table below, the nonlinear model learns circuits with similar connectivity patterns and therefore achieves comparable SHD values, supporting our claim that the linear formulation is sufficient for recovering the structural dependencies of interest.
>
> [4] Zheng, Xun, et al. "Learning sparse nonparametric dags." International conference on artificial intelligence and statistics. Pmlr, 2020.
>
> > Question 3: For anti-causal or ambiguous edges, could targeted activation patching or minimal interventional tests help refine causal direction without losing scalability?
>
> We agree that additional intervention-based validation on the circuits learned by CircuitLasso can effectively identify and correct anti-causal or spurious dependencies, as demonstrated by our single-prompt verification. Likewise, activation patching and other interventional tests can also be applied to our learned circuits. In terms of scalability, although such interventions incur similar computational costs as existing approaches like SHIFT, our framework can leverage coefficient-based measurements to prioritize only the most influential features or subcircuits. This substantially narrows the scope of required interventions and makes the verification process more efficient in practice.

---

> ### Author Response · Authors · 2025-11-25
>
> > Question 4: How robust are the learned circuits to the choice of SAE dictionary? Does retraining or altering the SAE dictionary significantly change the inferred circuit topology or key dependencies?
>
> We acknowledge that, similar to the existing circuit learning approaches on SAE features (Marks et al., 2025; Laptev et al., 2025a), our learned circuits are intrinsically subject to the specific version of SAE dictionary. Retrain or altering the SAE dictionary inevitably changes the permutation or disentanglement strategies of human-interpretable semantic meanings, hence results in a different set of SAE features and should, therefore, yield a different but structurally valid circuit. Moreover, the requirement for well-trained SAEs with low reconstruction error and effective disentanglement is not a limitation unique to CircuitLasso, but a prerequisite for feature-based mechanistic interpretability methods generally.  To address this field-wide concern and promote standardization, the community is actively committed to openly sharing high-quality, pre-trained SAEs (e.g., via platforms like Neuronpedia). The increasing availability of these validated, low-reconstruction-error SAEs allows different circuit-learning approaches to be benchmarked and compared fairly on a consistent and robust set of features.
>
> > Question 5: In addition, an important question concerns the practical scalability of the proposed framework given the high computational cost of training sparse autoencoders. CircuitLasso requires substantial data, compute, and hyperparameter tuning, which may limit the framework’s accessibility and generalizability to other models or domains. Could the authors comment on how the overall pipeline might be made more practical in multiple LLMs?
>
> The SAEs we use in our experiments, including those for GPT-2 Small, Pythia-70M, Gemma-2-2B, and Gemma-2-9B, are all open-sourced and widely adopted in prior works such as Meng et al. (2022) and Marks et al., (2025). As a result, **our method requires no SAE training cost**.
>
> Importantly, CircuitLasso can be directly applied to SAE features across consecutive layers for many LLMs. As demonstrated on the InterpBench dataset, our method achieves state-of-the-art accuracy in recovering circuit dependencies using exactly the same clean and corrupted clean of data used by intervention-based circuit-learning methods. Our theoretical complexity analysis and runtime comparison in Table 1 further highlight the **low computational cost** of our approach. Moreover, CircuitLasso has only **one hyperparameter**, the sparsity coefficient, in the regression objective. We provide an ablation study analyzing its effect on downstream prediction error and circuit sparsity, giving practical guidance for setting this parameter. Since our approach is data-efficient, computationally lightweight, and simple to implement, it can be readily applied to a wide range of existing open-source SAEs across diverse models and tasks.

---

> ### Author Response · Authors · 2025-11-25
>
> > Question 6: Experiments use a subset of 5 cases of the INTERPBENCH models for the main evaluation. However, the paper does not publicly specify the criteria used to select those specific cases from the full set of 86 (or the 16 main ones mentioned). Could the authors clarify how the subset was chosen and whether the selection introduces any bias in comparing circuit-discovery methods?
>
>
> We appreciate the reviewer’s suggestion to broaden the evaluation set. While our initial selection of 5 cases from the 85 synthetic ground-truth circuits was entirely random, we agree that results spanning a wider range provide a more complete picture of our method's performance. Per the reviewer’s request, we have now additionally evaluated our method on 12 more of the main cases, covering all 16 cases that are thoroughly benchmarked in the original Interpbench paper. These 16 cases together form a standard and widely adopted evaluation suite in this line of work and thus provide a standard and representative testbed for assessing circuit-learning approaches.
>
> We report the performance of the strongest baselines in terms of accuracy (EAP-ig) and efficiency (EAP), alongside our CircuitLasso (linear) and its nonlinear extension. We run each experiment three times and report the average SHD and runtime (rounded to the nearest integer) in the table. As shown in the table, the linear CircuitLasso achieves state-of-the-art accuracy with substantially better efficiency. We also observe that for cases whose ground-truth circuits are extremely sparse, typically involving only 1 edge (e.g., cases 20, 29–37), existing circuit-learning methods already achieve near-perfect accuracy (SHD = 2) with relatively low runtime (25–32 seconds). In these settings, our advantage is naturally less pronounced. However, because CircuitLasso allows better control over sparsity (via the sparsity coefficient and post-thresholding), it achieves slightly better accuracy on denser circuits. Overall, the broader results remain consistent with those of the 5 cases presented in Table 1: comparable or better accuracy and noticeably improved efficiency.
>
> We emphasize that our proposed approach is applicable to a wide range of models, datasets, and tasks. The evaluation on small-scale synthetic data serves as a sanity check under controlled conditions where the ground truth is available. This evaluation constitutes only a small portion of our full experimental scope. The same applies to our human-interpretable circuit examples and downstream task improvements using real-world data. There is, in principle, an unlimited number of downstream tasks to which such circuits can be applied. We believe that the breadth and depth of the experiments included in the paper provide strong and sufficient evidence of the effectiveness and generality of our approach.
>
> | Case |# Ground-truth Links| EAP SHD | EAP Runtime | EAP-ig SHD | EAP-ig Runtime | CircuitLasso-linear SHD | CircuitLasso-linear Runtime | CircuitLasso-nonlinear SHD | CircuitLasso-nonlinear Runtime |
> |------|------------|------------|---------------|------------|----------------|---------------------------|------------------------------|------------------------------|----------------------------------|
> | 13   | 22 |  12.3  |  37  |  9.3  |  42  |  6.7  |  **14**  |  **6.3**  |  77   |
> | 18   | 2 | 20.0  |  38  |  15.3  |  58  |  **11.0**   |  **20**  |  11.3  |  78  |
> | 19   | 10 |  11.3  |  32  |  9.3  |  40   |  8.0  |  **15**  |  **7.7**   |  83   |
> | 20   | 1 |  **2.0**  |  25  |  **2.0**  |  40  | **2.0** |  **15**  | **2.0** | 67 |
> | 21   | 17 |  23.0  |  39  | 21.7 | 57 | **11.0** | **13** | **11.0**  | 115 |
> | 26   | 2 |  5.3  |  32  | **3.7** | 40 | **3.7** | **20** | **3.7** | 76 |
> | 29   | 1 |  **2.0**  |  22  | **2.0** | 33 | **2.0** | **11** |  **2.0** | 52 |
> | 33   | 1 |  2.7  |  25  | **2.0** | 32 | 2.7 | **15** |  **2.0** | 55 |
> | 34   | 1 |  **2.0**  |  25  | **2.0** | 32 | **2.0** | **11** |  **2.0** | 55 |
> | 35   | 1 |  2.3  |  25  | **2.0** | 32 | 2.3 | **11** |  **2.0** | 53 |
> | 36   | 1 |  2.3  |  24  | **2.0** | 33 | **2.0** | **13** |  **2.0** | 53 |
> | 37   | 1 |  **2.0**  |  25  | **2.0** | 32 | **2.0** | **11** |  **2.0** | 50 |

---

### Official Review · Reviewer_mt7L · 2025-10-31

**Soundness:** 2
**Presentation:** 3
**Contribution:** 3
**Rating:** 2
**Confidence:** 4

**Summary:**

The authors introduce Circuit Lasso, a circuit discovery method based on Structured Equation Models. They evaluate circuits on Interpbench and SHIFT.

**Strengths:**

The authors provide a highly efficient approximation to circuit discovery, as measured by runtime.

**Weaknesses:**

1. The CircuitLasso method operates on X, a stacked activation vector, there is no weight-based analysis or backpropagation involved. The matrix A introduced in equation 1 therefore only reveals correlational information, and cannot make claims about causal dependencies.

2. The Circuit Discovery algorithms introduced by Marks et al. (SFC) and Conmy et al. (ACDC) are the main baselines for this work. This paper uses the SHIFT, a practical downstream application from SFC, for evaluation. This metric however only scores the identified set of relevant nodes and is invariant to the discovered set of edge weights.
   The Faithfulness and Completeness metric, used by both Marks et al. and Conmy et al. are a complementary signal to SHIFT, and would be a useful addition to this work. A gap in the existing literature is that SFC ablates sets of neurons, not edges. This contains the implicit assumption that a node in the circuit is connected to all subsequent nodes.
   A strong improvement over existing literature would be to evaluate faithfulness and completeness based on edge ablation. I would improve my score if this evaluation was run, and SFC results would be compared to Circuit Lasso results in line with SFC Figure 2.

**Questions:**

How large are the acquired circuits? Does the method involve setting a threshold to choose the top K attributed nodes / edges?

---

> ### Author Response · Authors · 2025-11-25
> **Responses to reviewer mt7L**
>
> We sincerely appreciate the reviewers’ thoughtful and constructive feedback. We have carefully addressed all concerns with additional analyses, experiments, and clarifications.
>
> **Clarification on CircuitLasso:** Our CircuitLasso approach efficiently recovers mechanistic circuits without requiring direct backpropagation or weight-based analysis on the original LLMs. Instead, all optimization (backpropagation and weighted-based analysis) is performed on the learned circuit itself. We show that circuits identified by prior, less efficient methods relying on heavy weight-based inspection and backpropagation through the full model can be accurately and more efficiently recovered by applying sparse linear regression to features extracted from the LLM, such as activation vectors or SAE features.
>
> **Causal Claims:** Our CircuitLasso approach is grounded in principles of SEM-based causal discovery. In SEM-based causal discovery, the matrix $A$ models relations among variables, and imposing an acyclic constraint on $A$ yields a directed acyclic graph learned from observational data. We leverage the established identifiability conditions (Peters et al., 2014) [1] by assuming causal sufficiency and specific noise characteristics (non-Gaussian or equal-variance Gaussian) to ensure the learned DAG is uniquely identifiable and thus interpretable as the underlying causal structure. We have included these discussions in the paper.
> - In particular, our method exploits the known, human-engineered computational order of the LLM. This architectural insight provides a justifiable acyclic constraint that aligns with the inherent feed-forward nature of the network (activations in later layers are computed after, and depend on, those in earlier layers).  Causal discovery with such a justified acyclicity leads to more accurate identification of the underlying causal relationships. On benchmark datasets Interpbench with known ground truth circuits, our approach achieves state-of-the-art graph learning accuracy and outperforms the majority of the existing circuit learning methods.
> - While the optimal acyclic constraint is less clear for SAE features, as their presumed monosemantic nature may conflict with the architectural computation order, our method still provides valuable mechanistic insights.  For instance, Figure 2 illustrates how a specific semantic concept propagates through the model and ultimately influences the target prediction. In addition, using our single-prompt intervention approach in lines 350-355 together with the multi-prompt approach in lines 345-348, we can systematically detect and refine the circuit to eliminate anti-causal or spurious correlations (e.g., 'ate/eat' leading to 'hunger'), thereby validating and improving the underlying sparse linear model. "
>
> [1] Peters, Jonas, et al. "Causal discovery with continuous additive noise models." The Journal of Machine Learning Research 15.1 (2014): 2009-2053.
>
> **Faithfulness and Completeness:** We appreciate the reviewer’s suggestion regarding the evaluation of edge importance as a complementary method to feature ablation. Since our paper already presents extensive experiments across multiple datasets, models, and SAEs, our goal here is to provide additional intuition about the learned circuit. We therefore assess its faithfulness and completeness of our learned circuit, on the CoLA dataset, following the evaluation protocol and implementation described in Marks et al. (2025).
> - Node Ablation: To validate the correctness of the evaluation process, we first apply the standard feature ablation approach to our learned circuit and compare against SHIFT. Let the learned circuit be $C$, and define the model output $m = p(Y=\text{grammatically correct}) - p(Y=\text{grammatically wrong})$. For fairness, we exclude SAE reconstruction errors and attention/MLP SAEs from SHIFT. For our CircuitLasso approach, we restrict attention to SAE features within the learned circuit, while ablating features in the original LLM. The top two plots in Figure 3 of the revised paper show the resulting node ablation curves. Our circuit achieves performance comparable to SHIFT, consistent with the observation in Marks et al. (2025) that relatively small feature circuits can explain a substantial fraction of a model’s behavior.
> - Edge Ablation: We then perform edge ablation on our circuit. Since our sparse-regression framework explicitly learns edge coefficients that capture the direct influence of each SAE feature on the output, we can ablate a given edge by zeroing its coefficient, thereby removing that dependency. We could not directly apply edge ablation to SHIFT because it does not provide edge-level correlations between SAE features and the output. The bottom two plots in Figure 3 of the revised paper report these results. The conclusions align with those from node ablation: a compact set of essential edges, and their associated SAE features, drives the model’s prediction behavior.

---

> ### Author Response · Authors · 2025-11-25
>
> > Question: How large are the acquired circuits? Does the method involve setting a threshold to choose the top K attributed nodes / edges?
>
> For CircuitLasso on neurons, the circuit is represented by an $A$ matrix of size $Ld \times Ld$. In practice, the learned $A$ is highly sparse. For example, in case 3 of the Interpbench dataset, $A$ has dimension $70 \times 70$, yet after thresholding we recover only $8$ causal edges. For CircuitLasso on SAE features, the circuit between consecutive layers is represented by $A_{i,j}$ of size $D \times D$. On the CoLA dataset with GPT2-small and pretrained SAEs, $A_{i,j}$ has dimension $32{,}768 \times 32{,}768$, and is likewise very sparse. As shown in Figure 3 in the Appendix, $83.3\%$ of edges have absolute weights below $0.1$ and $86\%$ below $0.41$. Thresholding at $0.5$ retains only about $10\%$ of the edges. While we do not threshold to select a subset of neurons, we recognize that applying a threshold to the **weighted adjacency matrices** for **edge selection** introduces an element of design choice. This thresholding is necessary to enforce the **sparsity requirement** essential for recovering an interpretable mechanistic circuit. We rigorously mitigate this potential heuristic by employing our downstream **single-prompt and multi-prompt validation strategies** (lines 345-355). These methods provide an **empirical and principled refinement mechanism** to confirm the functional necessity of the selected edges, thereby validating the resulting circuit structure.

---

### Author Response · Authors · 2025-12-03
**Summary**

We thank the reviewers for their thoughtful feedback and for recognizing the main strengths of our work. In particular, reviewers highlighted:

1. the novelty of formulating circuit learning as a sparse linear Lasso problem, which has not been explored in prior work (gmiW, mu8u, jgHf).

2. the efficiency and practical value of our continuous circuit learning approach, especially for high-dimensional human-interpretable SAE features (mt7L, gmiW, mu8u).

3. our comprehensive evaluation on both synthetic datasets with ground-truth circuits and real LLM tasks. The CoLA case study on GPT2-small was noted as a convincing demonstration of interpretable inner mechanisms (gmiW, jgHf).

**Key issues clarified during rebuttal**:

- We provided clarification on assumptions and technical details that were not fully captured in the initial reviews, including:
  1. Our causal sufficiency and identifiability assumptions and how they support causal interpretations of the learned graph.
  2. The rationale behind our computation order.
  3. The additional checks we performed to verify circuit accuracy (mt7L, gmiW, jgHf).

- We justified the use of a linear model by implementing a nonlinear variant and comparing performance on synthetic benchmarks with known circuits. The nonlinear version reduced efficiency while yielding comparable or slightly better accuracy, which supports our choice (gmiW, jgHf).

- We added comprehensive evaluation using the faithfulness and completeness metrics that are standard in prior work (mt7L).

- We also clarified our decision not to incorporate token position awareness (mu8u).

We believe that we have fully addressed all concerns and that the revisions have substantially strengthened the paper. We appreciate the Area Chairs for their time, consideration, and the granted extension. We hope that the Area Chair will find that the revised submission adequately responds to the reviewers’ feedback and supports a positive decision.

---

### Meta-Review · Area_Chair_5M96 · 2025-12-31

**Summary:**

The core concerns are two key limitations of the proposed CircuitLasso method. First, reviewers challenged the rigor of the paper’s causal claims: the method uses observational data and linear assumptions, with no adequate quantitative interventional experiments to tell causal relationships apart from correlations. Second, the method relies on pre-trained sparse autoencoders (SAEs), which limits its generalizability to models without existing SAE resources. Therefore, I would recommend rejection.

**Reviewer Concerns:**

The authors partially addressed minor reviewer concerns, including clarifying SEM-based causal assumptions, adding faithfulness/completeness evaluations via edge ablation, expanding Interpbench evaluations to 16 cases, and discussing nonlinear extensions. However, key outstanding concerns persist. However, there still remain unresolved issues, including the lack of rigorous quantitative experiments proving causal relationships (as the proposed single/multi-prompt methods are insufficient for definitive causal validation) and the limited generalizability due to SAE dependence, authors only note existing pre-trained SAEs but fail to address applicability to models without such resources.

**Reviewer Scores:**

Reviewer mt7L may potentially increase the score given the added faithfullness/completeness experiments. The remaining reviewers may keep their scores given their reviews and the authors' response.

---

### Decision · Program_Chairs · 2026-01-26

Reject